# Cascades of effectiveness of next-generation insecticide-treated nets against malaria, from entomological trials to real-life conditions

Clara Champagne [1,2] ✉, Jeanne Lemant [1,2], Alphonce Assenga [3], Ummi A. Kibondo [3], Ruth G. Lekundayo[3], Emmanuel Mbuba[1,2,3], Jason Moore[1,2,3], Joseph B. Muganga[3], Watson S. Ntabaliba[3], Olukayode G. Odufuwa[1,2,3], Johnson Kyeba Swai[1,2,3], Maria Alexa [1,2], Roland Goers[1,2], Monica Golumbeanu [1,2], Nakul Chitnis [1,2], Amanda Ross[1,2], Raphael N'Guessan[4,5], Sarah Moore [1,2,3] & Emilie Pothin[1,2,6]

As insecticide resistance spreads in Africa, next-generation insecticide-treated nets (ITNs) are increasingly being deployed to protect vulnerable populations against malaria. While these nets provide greater entomological efficacy against resistant mosquitoes, their effectiveness against malaria transmission also depends on other factors, such as durability, access, usage, and activity patterns of hosts and vectors. Here, we quantify the impact of two next-generation ITNs, namely Interceptor®G2 (chlorfenapyr-pyrethroid) and Olyset® Plus (piperonyl butoxide-pyrethroid), in a cascade from entomological efficacy to population-level effectiveness. We use a mathematical model that we parameterize with entomological data and validate against results from randomized controlled trials. We found that, beyond entomological factors, operational factors including functional survival, ITN use and in-bed exposure critically impact ITN effectiveness overall and per ITN types. Our results obtained for Tanzania can be extended to other contexts in a dashboard allowing users to explore product selection based on setting-specific factors that influence ITN effectiveness.

Insecticide-treated nets (ITNs) are the cornerstone of malaria control and are recommended by the World Health Organization (WHO) in all malaria-endemic geographies[1]. Since 2004, over three billion ITNs have been delivered[2] and between 2000 and 2022 ITNs have averted an estimated 1.5 billion malaria cases globally[3,4]. ITNs protect individuals against malaria through three mechanisms. Firstly, they act as a physical barrier, preventing mosquitoes from biting hosts sleeping under the net. Secondly, the insecticide interferes with mosquito host-seeking and prevents bites even when the ITN has become damaged. Thirdly, mosquitoes that come into contact with the net may be killed by the insecticide or suffer non-lethal adverse effects, such as reduced fertility. Therefore, ITNs provide individual protection for the net user and community protection through shorter mosquito survival, reduced mosquito populations and a reduced infectious reservoir[5,6].

[1]Swiss Tropical and Public Health Institute, Allschwil, Switzerland. [2]University of Basel, Basel, Switzerland. [3]Ifakara Health Institute, Bagamoyo, Tanzania. [4]London School of Hygiene and Tropical Medicine, Department of Disease Control, London, UK. [5]Vector Control Product Evaluation Centre, Institut Pierre Richet, Bouaké, Côte d'Ivoire. [6]Clinton Health Access Initiative, Boston, MA, USA. ✉e-mail: clara.champagne@swisstph.ch

Standard ITNs contain insecticides from the pyrethroid class, such as alpha-cypermethrin, deltamethrin or permethrin. However malaria vector mosquitoes have become increasingly resistant to pyrethroids, especially on the African continent and the insecticidal efficacy has substantially decreased[7]. To counteract this, next-generation ITNs containing additional active ingredients combined with pyrethroids have been developed. Several have been recommended by the WHO based on randomised controlled trials (RCTs) that demonstrated public health impact[1] followed by prequalification[8], after their quality, safety and entomological efficacy were demonstrated through experimental hut trials (EHT).

The first class of next-generation ITNs to be pre-qualified contain pyrethroids combined with piperonyl butoxide (PBO), a synergist that restores pyrethroid sensitivity in resistant mosquitoes by inhibiting detoxifying enzymes[9]. In 2023, the WHO also recommended ITNs in which pyrethroids are combined with chlorfenapyr. Chlorfenapyr is a pro-insecticide activated only after metabolism within the target insect[10]. Once ingested or absorbed, detoxifying enzymes—often upregulated in pyrethroid-resistant mosquitoes—metabolise it into the active compound tralopyril, which disrupts cellular energy production. The improved effectiveness of PBO and chlorfenapyr nets relative to standard pyrethroid ITNs was shown in RCTs and pilot studies[11-15] and in 2023, PBO nets represented more than half of the ITNs distributed in sub-Saharan Africa and dual active ingredient ITNs, which include chlorfenapyr nets, about 20%[4].

Unfortunately, insecticide resistance is not the only factor reducing the effectiveness of ITNs. Additional considerations, including population net access, usage and durability (net being physically intact and insecticidally active) are key to guarantee ITN effectiveness for malaria control[16]. In 2023, overall access to ITNs was estimated at around 60% in Africa, with only five of the 40 highest-burden countries on the continent achieving the target of 80% of the population sleeping under an ITN[4,17]. With national ITN campaigns usually planned to happen every 3 years, it is crucial that nets are retained and maintain their efficacy over this time frame. In practice, much shorter ITN survivorship has been observed in many settings with an estimated overall median retention time of 1.64 years among all countries in Africa[17]. Finally nets only protect hosts when they are under the net and, depending on both mosquito and humans' activity patterns, an appreciable proportion of vector biting may take place while individuals are unprotected outside of sleeping hours[18-22]. For nets to be effective at both individual and population level, they have to be present and in regular use, in good physical condition and remain insecticidal, thereby providing protection through preventing bites and killing disease vectors for the population at risk of malaria.

Various indicators exist to monitor insecticide resistance, ITN use, ITN durability and in-bed exposure (see Box 1) but these indicators are usually measured independently of each other and therefore do not provide information on the relative contribution of each factor to the overall public health impact of ITNs. Quantifying this relative contribution is important for public health, as it helps stakeholders identify context-specific factors that reduce ITN effectiveness and prioritise strategies to address them.

The contribution of these different factors on the pathway from efficacy to effectiveness can be explored using the concept of effectiveness cascade[23]. Such cascades have often been used to represent the impact of health systems factors on malaria case management, both conceptually[24-26] and quantitatively[27,28]. For vector control and ITNs in particular, while conceptual cascades have been presented in the literature[23,29], they have rarely been formally quantified[30-32] due to the difficulty of disentangling factors that are intrinsically intertwined in real-life settings. Mathematical models of malaria transmission offer the possibility to disentangle each factor and quantify their relative contribution to the overall ITN effectiveness. Two previous modelling studies (Briet et al.[33] and Bertozzi-Villa et al.[17], including the related analysis presented in the 2022 World Malaria report (https://www.who.int/teams/global-malaria-programme/reports/world-malaria-report-2022)) explored this topic. Both studies highlighted the critical importance of operational factors, such as access, use and durability. However except for one PBO-pyrethroid ITN included in ref. 33, both studies focused on pyrethroid-only ITNs and neither study directly incorporated human and vector activity rhythms in their quantifications. Additionally, both studies identified a substantial variability across countries and settings. There is therefore a need for additional tools that can explicitly quantify the relative contributions of the factors affecting the effectiveness of next-generation ITNs in various contexts. This would enable decision-makers to assess which products are likely to perform best in their specific settings.

With mathematical models of malaria transmission, epidemiological conditions can be reproduced in silico and the impact of specific control interventions on risk and burden can be quantified in areas or configurations in which they have not formally been tested through

---

## BOX 1:
# Definitions

**Entomological efficacy:** capacity of a given vector control product to reduce mosquito transmission potential, through the alteration of mosquito biting, mortality and/or fecundity, under ideal operational conditions. It can be quantified as the relative reduction in vectorial capacity through deploying the vector control product under ideal operational conditions.

**ITN effectiveness:** capacity of a given ITN product to reduce mosquito transmission potential under user conditions. It differs from entomological efficacy in that insecticidal durability, functional survival, ITN use and in-bed exposure are also accounted for. It can be quantified as the relative reduction in vectorial capacity through deploying the ITN under realistic operational conditions.

**Vectorial capacity:** Total number of potentially infectious bites originating from all the mosquitoes biting a single perfectly infectious (i.e. all mosquito bites result in infection) human on a single day. It represents the potential for a given mosquito population to transmit malaria[68].

**ITN use:** Proportion of the population sleeping under a net the previous night. It is the result of both access to a net (proportion of the population with access to an ITN for every two persons that slept in the household the previous night) and use given ownership (proportion of the population with ITNs that slept under a net the previous night).

**Insecticidal durability:** "Retention of insecticidal efficacy of the ITN"[69].

**Functional survival:** "An estimate of the physical lifespan of an ITN product in the field [...] It is measured as an ITN that is present in use and in serviceable condition"[69]. It is the result of the ITN's physical integrity declining due to damage and attrition that results in nets being discarded when the user no longer regards them as useful.

**In-bed exposure:** The proportion of vector bites occurring indoors during sleeping hours out of all potential bites in day (indoors and outdoors), for an unprotected individual. It represents "the maximum possible personal protection any intervention targeting sleeping spaces could provide"[70].

trials[34–36]. Therefore, these models enable us to extract further information from the entomological and epidemiological trials, once they have been calibrated and validated against the data from these trials. In particular, EHT data have been used to estimate the entomological efficacy of various vector control tools and hence parameterise mathematical models of malaria[37–42]. Population-level RCTs can be used as validation sets to evaluate the capacity of the model to predict epidemiological outcomes accurately[36,40]. Regarding ITNs specifically, parameterised and validated models have been developed for pyrethroid and PBO ITNs[37,40] as well as chlorfenapyr ITNs[36], but they have not been used to quantify cascades of ITN effectiveness.

In this work, we quantify the epidemiological impact of two next-generation ITNs, namely Olyset® Plus (PBO-pyrethroid) and Interceptor® G2 (chlorfenapyr-pyrethroid). First, the entomological efficacy of the ITNs is estimated using a Bayesian statistical model fitted to data from eight EHT conducted in Tanzania, Côte d'Ivoire and Benin. Secondly, these entomological efficacy estimates are incorporated into an established mathematical model of malaria transmission (the Open-Malaria simulation platform[43]) and the model's capacity to predict population-level prevalence is validated against the outcomes of four RCTs conducted in Tanzania, Benin and Uganda. Finally we use the mathematical model to quantify the impact of both net types along the different steps of the effectiveness cascade and identify the relative contributions of the main factors responsible for the loss of effectiveness.

## Results

### Estimation of next-generation ITN entomological efficacy from experimental hut trial data

With a Bayesian model adapted from Denz et al.[38] (see 'Methods' section) we estimated four indicators: reduction in host availability (reflecting the deterrent properties of the nets), pre-prandial killing effect (referring to mosquito mortality before biting), post-prandial killing effect (referring to mosquito mortality after biting) and overall entomological efficacy (measured as the reduction in vectorial capacity and is a combination of the previous three indicators). The indicators were estimated for Interceptor® G2, Olyset® Plus and five pyrethroid-only ITNs (Interceptor®, MiraNet®, MAGNet®, Olyset®, PermaNet® 2.0) using data from eight EHTs conducted in Tanzania[44–47], Côte d'Ivoire[44] and Benin[48,49] (cf. Fig. 1, with all numerical outputs in Supplementary Tables 1.1–1.4). On average, the reduction in host availability was estimated to be higher for Olyset® Plus compared to Interceptor® G2 and all pyrethroid-only ITNs tested. In contrast, the pre-prandial killing effect was highest for Interceptor® G2. Values obtained for Olyset® Plus were lower on average, but substantially higher than those for pyrethroid-only ITNs. Post-prandial killing effects were more uncertain and displayed stronger variations across trials, even though both next generation ITNs displayed higher killing than pyrethroid-only ITNs. This uncertainty can be explained because in many trials very few fed mosquitoes were collected (see Supplementary Tables 2.1 and 2.2), thus the proportion of killed among fed mosquitoes was estimated from very few observations. Overall, for all three indicators, there was also substantial heterogeneity across EHTs, as well as uncertainty within trials reflected in the boxplots (Fig. 1) and by the size of the estimated credible intervals (Supplementary Tables 1.1–1.4).

Combining all three previous indicators, the entomological efficacy, measured in terms of vectorial capacity reduction, was on average 95% for unwashed Interceptor® G2, 92% for unwashed Olyset® Plus and 82% for unwashed pyrethroid-only ITNs. In all studies but two (Assenga et al.[44], Tanzania data and Martin et al.[47] for *An. funestus* mosquitoes), the difference in vectorial capacity reduction between next-generation ITNs and pyrethroid-only ITNs was positive in more than 95% of the simulations, indicating the superiority of next-generation ITNs in terms of entomological efficacy (see Supplementary Table 3). In the studies which included both types of next-

generation net within the same trial in Tanzania (Assenga et al.[44] and Martin et al.[47]), the difference in vectorial capacity reduction estimates between Interceptor® G2 and Olyset® Plus had a 95% credible interval containing the value 0. This indicates that, considering the statistical uncertainty, we could not conclude that there was a difference in impact between these two products. In the EHT conducted in Côte d'Ivoire[44], however the interval did not contain 0 and a higher entomological efficacy was measured for Interceptor® G2.

Six EHTs include arms in which ITNs have been artificially aged (washed 20 times and deliberately damaged), in order to simulate what the efficacy would likely be after 3 years of operational use[8]. The other two EHT also tested operationally-aged ITNs (ITNs retrieved from people's homes) in separate arms[47,49] For Interceptor® G2 and Olyset® Plus, vectorial capacity reduction was consistently higher for unwashed/new ITNs compared to washed/operationally-aged ones (except for the BIT080 trial and the Assenga et al. trial in Côte d'Ivoire[44] where both values were identical). Interestingly, in some instances, an increase in deterrence or post-prandial killing effect was observed, but this was compensated by the decrease in pre-prandial killing efficacy, such that the overall vectorial capacity reduction decreased compared to unwashed nets. Overall, the decrease in entomological efficacy due to washing was small in most trials, below 3% reduction for Interceptor® G2 and below 7% for Olyset® Plus. However the decrease in entomological efficacy between new and operationally-aged ITNs was very large in the Martin et al.[47] trial, with a decrease of 16 and 33% across ITN products and vector species, suggesting additional insecticide loss through evaporation or photodegradation. It was lower in the Sovegnon et al.[49] trial (3% decrease). For pyrethroid-only ITNs, vectorial capacity estimates were more variable across trials and washed/operationally-aged nets did not systematically display a lower entomological efficacy compared to unwashed ones.

### Model validation: reproduction in silico of randomized control trial data

The aforementioned estimates (outputs displayed in Fig. 1), alongside setting-specific information related to epidemiological conditions and ITN effectiveness, were then included in the OpenMalaria mathematical model of malaria transmission[43]. To assess the capacity of this parameterisation of OpenMalaria to quantify the epidemiological impact of ITN deployments, four RCTs conducted in Tanzania (Mosha et al.[11,12] and Protopopoff et al.[50,51]), Benin (Accrombessi et al.[13,14]) and Uganda (Staedke et al.[52]) were reproduced in silico (cf. Fig. 2). Open-Malaria was calibrated to reproduce the pre-intervention prevalence in each arm and the post-intervention prevalence estimates were used for validation only. Overall trials, the uncertainty intervals for the model predictions and observations overlapped at all time points for Interceptor® G2 and all time points but two for Olyset® Plus. In the trial by Staedke et al.[52], the second observation was slightly missed. In the trial by Protopopoff et al.[50,51], the observation at 28 months in the intervention arm was largely missed; however the observed prevalence at that date was abnormally high in both the control and intervention arm, the estimates being higher than all other observations, including the pre-intervention estimate. For pyrethroid-only ITNs, although the models did reproduce general trends, seven out of 17 observed prevalence points were not captured by the model estimates. In Mosha et al.[11,12], two observations were slightly missed. In Protopopoff et al.[50,51], three observations with high prevalence toward the end of the trial were missed, in particular the observations at 21 and 28 months. In Accrombessi et al.[13,14], the impact of pyrethroid-only ITNs was underestimated at the beginning and at the end of the post-intervention period.

Nonetheless, there was a general agreement between observations and model estimates (see Fig. 3): in the linear regression between observed and modelled values, the adjusted $R^2$ was 0.94 for prevalence, 0.64 for effect sizes and 0.56 for relative comparison to pre-

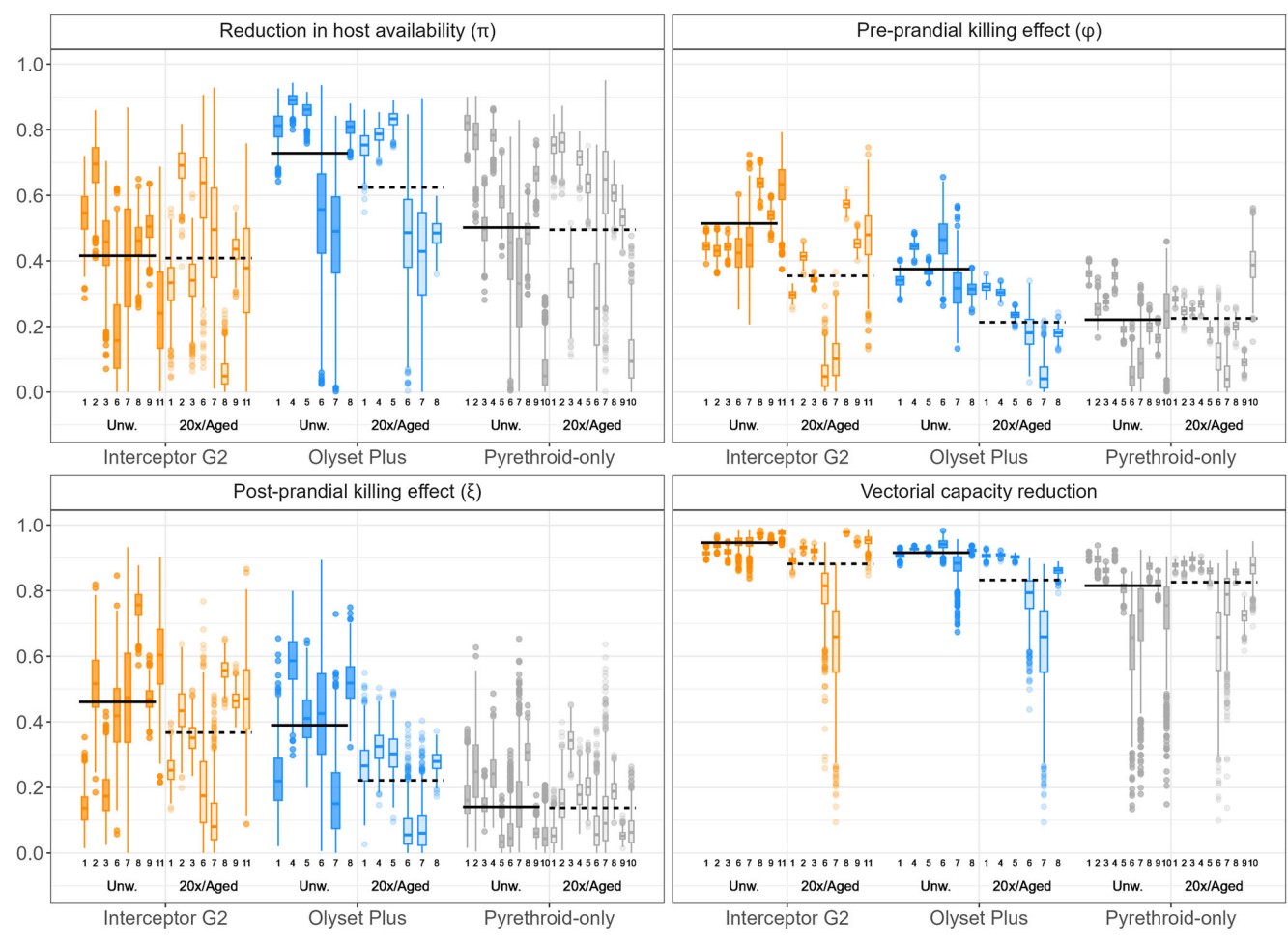

**Fig. 1 | Estimates of entomological efficacy for unwashed and washed/aged nets compared to untreated nets as control.** The black horizontal lines represent the average for unwashed nets (solid line) and 20-times washed/aged nets (dashed lines) across all experimental hut trials (EHT) per net type. Each boxplot corresponds to a specific EHT: (1) Assenga et al. (Tanzania)[44], (2) Kibondo et al. (Tanzania)[45], (3) BIT080 (Tanzania), (4) BIT055 (Tanzania), (5) Odufuwa et al. (Tanzania)[46], (6) Martin et al. (*An. gambiae*, Tanzania)[47], (7) Martin et al. (*An. funestus*, Tanzania)[47], (8) Assenga et al. (Côte d'Ivoire)[44], (9) Nguessan et al. (Benin)[48],

(10) Sovegnon et al. (trial 1, Benin)[49], (11) Sovegnon et al. (trial 2, Benin)[49]. The data are derived from 1000 samples from the posterior distribution of each parameter. In each boxplot, the central line represents the median, the lower and upper hinges represent the 25th and 75th percentiles and the whiskers extend to the largest and smallest values within 1.5 times the interquartile range from the hinges. Data points beyond the whiskers are plotted individually. 'Unw' refers to unwashed nets (depicted darker boxplot shading) and '20×/aged' refers to nets washed 20-times/ aged (depicted with lighter boxplot shading).

intervention levels. The regression coefficients for each indicator were 0.98, 1.18 and 0.80, respectively.

### Cascades of ITN effectiveness

Finally we used the validated model to measure the decline in effectiveness between entomological efficacy and population-level conditions and quantified the relative contribution of five factors to this decline: entomological efficacy, functional survival, usage at distribution, insecticidal survival and in-bed exposure (Fig. 4). Effectiveness was measured in terms of vectorial capacity reduction based on all EHT trials conducted in Tanzania and population-level conditions are taken to represent the trial by Mosha et al.[11,12], but assumptions can be modified to reflect other settings in an interactive dashboard: https://swisstph.shinyapps.io/ITNcascadesDashboard/.

While the effectiveness of both net types was above 90% when only entomological efficacy was considered, it dropped below 60% when all other dimensions were included, highlighting the important contribution of operational factors beyond ITNs entomological efficacy (killing and feeding inhibition).

The overall effectiveness of Interceptor® G2 nets was higher than the one of Olyset® Plus nets, reducing vectorial capacity on average by 59% over 3 years, against 45% for Olyset® Plus. The largest difference in effectiveness between the two ITNs came from functional survival: this factor was responsible on average for a drop of 5 percentage points for the Interceptor® G2 against 19 for the Olyset® Plus. On the contrary, when results from all EHTs were averaged, the overall differences in entomological efficacy between the two ITNs were minimal (cascades per EHT are displayed in Supplementary Fig. 4 and in the interactive dashboard).

Beyond entomological efficacy, the main factors responsible for the decline in effectiveness for both ITNs were imperfect usage at distribution (responsible for a drop of 11 points) and in-bed exposure (responsible for a drop of 13–14 points, with differences across net types that could be attributed to statistical uncertainties and non-linearities in vectorial capacity calculations). On the other hand, insecticidal durability had a smaller role in the loss of effectiveness (responsible for a drop of 3–5 points). However this was very variable depending whether insecticidal durability is measured after 20 washes (a drop of 0–1 point) or after ageing in the field (a drop of 7 –12 points).

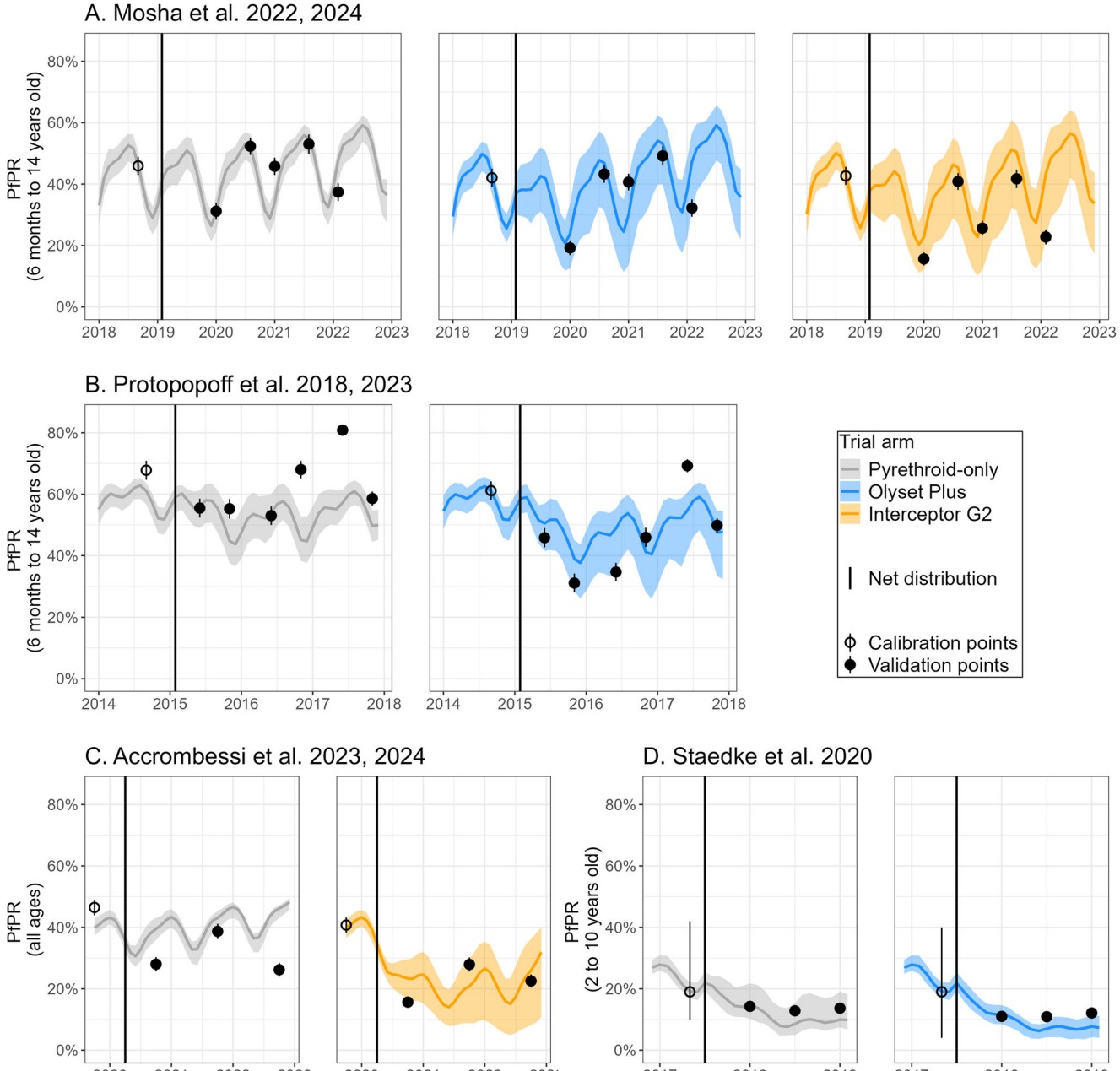

**Fig. 2 | Validation of model Plasmodium falciparum prevalence (PfPR) estimates (curve and shaded areas) against measurements from the RCTs (dots).** Plain line represent the median model prediction across EHT trials (7 trials in East Africa, 3 in West Africa), transmission intensity calibration and model stochasticity (10 stochastic replicates). Shaded areas represent the minimum and maximum over these uncertainty ranges (see 'Methods' section for methodological details). Plain dots indicate observations that were not used to calibrate the model and are solely presented for validation. White dots were used in the calibration process. For each dot, the point estimate corresponds to the raw proportions reported in each trial (all computed over more than 800 individuals) and error lines represent 2 empirical standard deviations from the point estimate. The black vertical line indicates the ITN distribution. **A** Trial by Mosha et al.[11,12] **B** Trial by Protopopoff et al.[50,51] **C** Trial by Accrombessi et al.[13,14] **D** Trial by Staedke et al.[52]

These results are robust to changes in the ordering of the cascade: the relative importance of the four factors remained broadly similar, with changes in the loss in percentage points in the same order of magnitude as the Bayesian credible intervals in Figure 4 and the ordering of the factors remaining generally the same (see Supplementary Tables 5.1 and 5.2).

## Discussion

In this work, we quantify the impact of two next-generation ITNs, namely Interceptor® G2 (chlorfenapyr-pyrethroid) and Olyset® Plus (PBO-pyrethroid), in a cascade from entomological efficacy to population-level effectiveness. We found that, beyond entomological factors, operational factors including functional survival, initial ITN use and in-bed exposure play a crucial role in ITN impact, both overall and when comparing ITN types. Our results obtained for Tanzania can be extended to other settings using an interactive dashboard (https://swisstph.shinyapps.io/ITNcascadesdashboard). The dashboard allows users to explore various combinations of activity rhythms, durability properties, ITN use and mosquito species. These cascades make it possible to explore the main bottlenecks reducing the effectiveness of ITNs at the population level. During the RCT conducted by Mosha et al.[11,12], functional survival was highlighted as a substantial

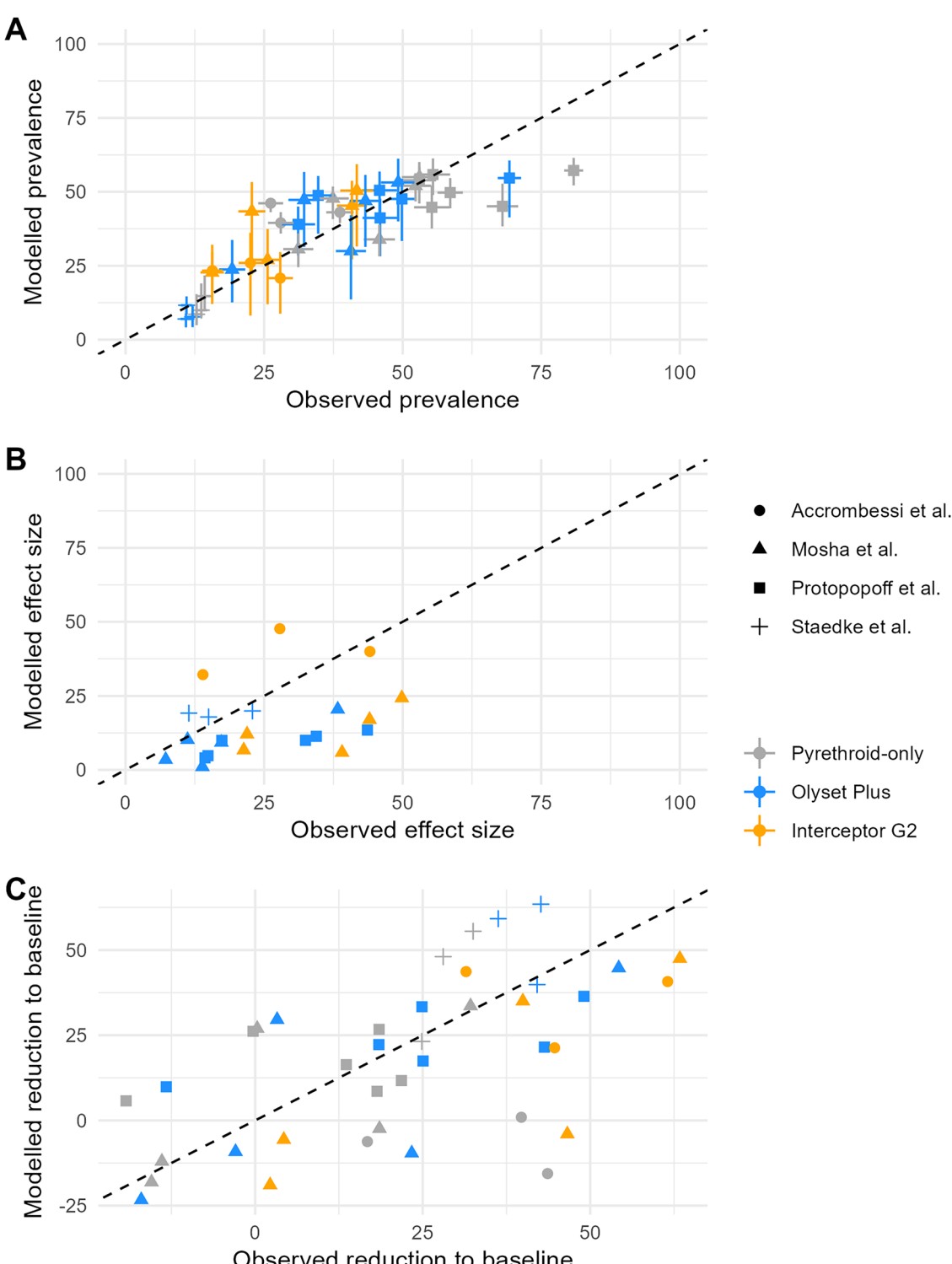

**Fig. 3 | Comparison of observed values in RCTs (x-axis) and modelled estimates (y-axis), considering validation points only.** The black dashed line indicates the Y = X line, i.e. perfect agreement. **A** Observed and modelled prevalence, with associated uncertainty intervals. On the y-axis, point estimates represent the median model prediction across EHT trials (7 trials in East Africa, 3 in West Africa), transmission intensity calibration and model stochasticity (10 stochastic replicates). Error lines represent the minimum and maximum over these uncertainty ranges (see 'Methods' section for methodological details). On the x-axis, the point estimate correspond to the raw proportions reported in each trial (all computed over more than 800 individuals) and error lines represent 2 empirical standard

deviations from the point estimate. **B** Observed and modelled effect sizes (relying on the median model prediction across EHT trials, transmission intensity calibration and model stochasticity). Effect size is defined as $100 \times \frac{PR_0 - PR_i}{PR_0}$, where $PR_0$ is the prevalence in the control arm and $PR_i$ is the prevalence in the intervention arm. **C** Observed and modelled relative change compared to baseline (relying on the median model prediction across EHT trials, transmission intensity calibration and model stochasticity). Relative change compared to baseline is defined as $100 \times \frac{PR_{t=0} - PR_t}{PR_{t=0}}$, where $PR_{t=0}$ is the prevalence at baseline (pre-intervention) and $PR_t$ is the prevalence post-intervention.

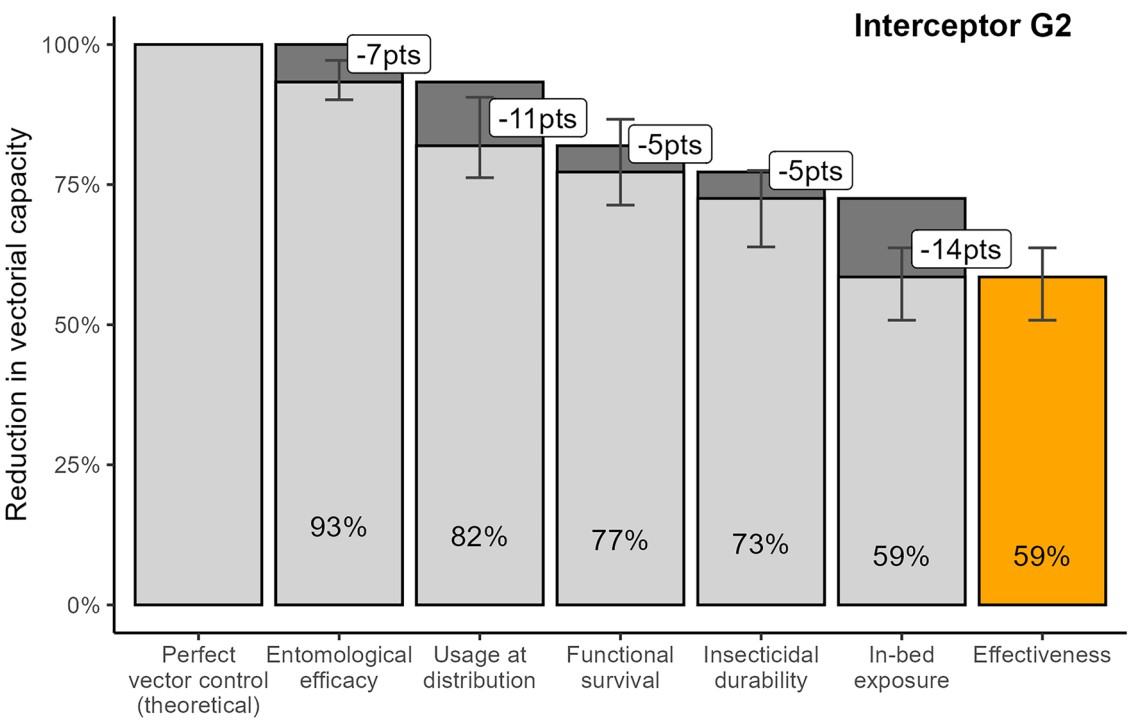

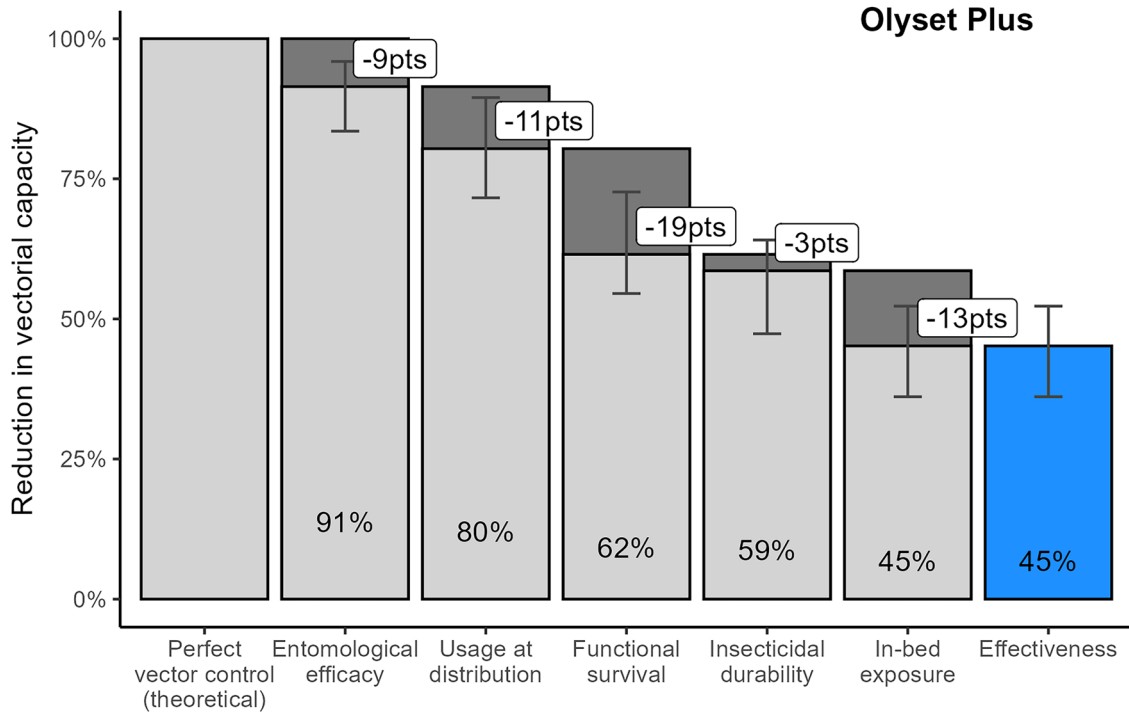

**Fig. 4 | Cascades of ITN effectiveness, reflecting average entomological efficacy estimates from all seven EHTs conducted in Tanzania and setting-specific conditions of the Mosha et al.[11,12] randomized controlled trial.** Top panel (orange): interceptor® G2. Bottom panel (blue): Olyset® Plus. Bars represent the mean and error bars represent the 95% credible intervals from EHT fitting of entomological efficacy, based on 1000 samples from each EHT and taking the ensemble over all EHT for each net type (see 'Methods' section). Cascades for each EHT individually are displayed in Supplementary Fig. 4.

contributor to effectiveness decay for Olyset® Plus, while imperfect usage and exposure outside sleeping hours were important for both net types. Such considerations provide valuable information for decision-makers who face resource-constrained choices, such as transitioning to next-generation nets, improving deployment strategies, improving ITN access between mass distribution campaigns, conducting behaviour change communication campaigns or

introducing supplementary vector control interventions. Although all these interventions are valuable, understanding their relative importance can help focus efforts on the most critical factors for maximising malaria control in a given setting.

Our model is based on estimates of entomological efficacy inferred from eight EHT conducted in Tanzania, Benin and Côte d'Ivoire. We found that Interceptor® G2 presents strong pre-prandial killing

properties while Olyset® Plus has a higher deterrence effect compared to pyrethroid-only nets. Substantial variation was observed across EHT, especially among the estimates for pyrethroid-only ITNs. Some of the variability for pyrethroid-only ITNs may come from differences between products (e.g. nets treated with permethrin, deltamethrin or alpha-cypermethrin that vary in terms of their irritancy to mosquitoes) or differences in resistance levels across study sites. In entomological trials the choice of active comparator is a critical determinant of relative ITN performance[53]. It is often assumed that all pyrethroids behave in the same way, but there is abundant evidence that their mode of action varies from quite high irritancy to very low irritancy[54]. Additionally, EHT data often present large intrinsic variability in mosquito numbers and mortality[9,42] that could contribute to these variations. Our statistical model to infer entomological efficacy estimates from EHT data is general and could be readily applied to other EHT datasets beyond the eight studies presented here, to better understand the drivers of entomological efficacy and the variability between trials[9,41,42].

At the population level, we validated our dynamical model's ability to reproduce the reduction in malaria prevalence observed in four RCTs conducted in Tanzania, Uganda and Benin. These trials consistently showed greater effectiveness of next-generation ITNs compared to pyrethroid-only nets, though the duration of protection varied substantially across net types and settings. Despite some discrepancies, our model reproduced the overall magnitude of the impact of next-generation ITNs on prevalence and captured much of the associated uncertainty. When considering the effect sizes between next-generation and pyrethroid-only ITNs and the relative comparison between pre- and post-intervention levels, substantial variability was observed in both the observed and modelled values within and across trials. Although the model did not capture perfectly the observed heterogeneity for these two indicators, there was agreement in the general trends. Overall, this evidence from RCTs is in line with the results observed the effectiveness pilots conducted in Burkina Faso, Mozambique and Rwanda, that also observed a superior effectiveness of chlorfenapyr and PBO nets relative to pyrethroid ones for at least 2 years after distribution[15]. Less clear results were observed in Nigeria where diverging trends in prevalence and incidence could be attributed to data quality issues, confounding interventions and external factors and where ITNs usage was low[15]. Therefore, our updated parameterisation of the OpenMalaria model, which has been tested against RCT data and demonstrates promising alignment especially for prevalence outcomes, can now be used to predict potential ITN impact in places where there have not been RCTs. It can also be used for subnational tailoring of interventions and/or country-wide predictions of the impact of malaria interventions[35,55].

Our analysis sheds light on the differences in impact between the entomological and the population levels, by identifying the main contributors of ITN effectiveness beyond entomological efficacy. Functional survival was highlighted as the main factor responsible for the loss of effectiveness of the studied PBO net in Mosha et al.[11,12]. This was not so pronounced for Interceptor® G2 in that trial, but ITN survival can vary widely across and within countries due to differences in the net use environment and user behaviour[33]. A large decrease in Interceptor® G2 use in the third year post-distribution was observed in another trial conducted in Benin that was sufficient to reduce impact on malaria[14]. Functional survival is dependent on both human behaviour and fabric integrity as nets are discarded by their users when they are perceived as too torn even if they are still effective in killing mosquitoes[33,56]. We estimated functional survival from survey data on ITN usage among trial participants but longitudinal follow-up of the nets in the same setting revealed lower functional survival estimates[57]. In RCTs, frequent hang-up campaigns, follow-up visits and community sensitisation efforts help maximise ITN access and use. Under real-world conditions, however populations typically receive new ITNs only every 3–4 years, without supplemental distributions or regular

net-care messaging, so functional survival is expected to be lower[17]. Therefore, lower functional survival would contribute even more to the loss of effectiveness. In practice, for other settings, online tools (e.g. https://allianceformalariaprevention.com/itn-quantification) are available to estimate functional survival and ITN needs over time using survey data and stock-and-flow models[17,58].

ITN use and access at the time of distribution was also identified as a key contributor to ITN impact, despite the high values observed in the RCTs (around 70% or above). These high values are likely due to the controlled conditions of the RCT, where net access is guaranteed and trial sensitization may promote high usage among those who own a net. In real-world settings, however the proportion of people sleeping under a net is typically lower—even immediately after distribution[59]—primarily due to limited access to ITNs and this remains a major bottleneck to ITN effectiveness.

Activity rhythms of humans and mosquitoes were also highlighted as a substantial contributor of ITN effectiveness. This is specific to the Tanzanian dataset on human and mosquito activity patterns used in the model, in which human individuals wake up very early, while mosquitoes are still active. Additional representative data on vector and population behaviours would be important to further quantify this impact[18,21]. Overall, this aspect should not be overlooked and tools that address human-vector contact outside of sleeping hours may have a role to play in reducing the gaps in the effectiveness cascade.

The impact of insecticidal durability depended on the approach used to represent net ageing. Operationally-aged ITNs were included in only two EHTs while in other EHTs, insecticidal durability was quantified comparing unwashed and 20-times washed nets as a proxy for operational-ageing. There is recent evidence that washing may not realistically demonstrate insecticide loss under user conditions for PBO or chlorfenapyr nets[47,60]: the effect of insecticidal durability on overall effectiveness may therefore be underestimated when based on ITN washing only because additional insecticide may be lost through evaporative loss[61], the formation of large crystals (blooming) or damage when nets are exposed to sunlight. The stark difference in entomological efficacy between artificially aged (20× washed) and operationally aged (used for several years) ITNs underscores the need for community studies or post-market surveillance to accurately assess product effectiveness, as well as the development of better proxies for operationally aged ITNs in EHT.

Even though our model has been validated to reproduce four controlled settings and effectiveness cascades were presented for Tanzania, the predicted effectiveness cascade in other contexts can be explored with our interactive dashboard. Ecological or behavioural characteristics, such as mosquito species and activity rhythms can be modified to represent other geographies. Intervention characteristics, such as ITN access or usage and durability (median functional survival), can also be modified to represent operational settings beyond the optimal deployment conditions observed in RCTs. Therefore, this online interactive tool can enable malaria programmes and stakeholders to explore the cascades using setting-specific assumptions relevant to their context.

This work has nonetheless some limitations. First, we estimated ITN entomological impact using EHTs from three countries: in Tanzania, results were obtained for *Anopheles arabiensis* in five trials and *Anopheles funestus* in one trial, in Benin and Côte d'Ivoire, the results were obtained for *Anopheles gambiae s.l.* in three trials. Therefore, caution should be exercised when extrapolating to other countries or vector species. Additionally, our estimates of post-prandial killing efficacy of the different net types displayed large uncertainties, due to small sample sizes among fed mosquitoes in these EHT. These uncertainties are nonetheless quantified through our statistical model and propagated into the effectiveness cascades.

Second, despite an overall good agreement between observations and model predictions, there were divergences in some instances, notably in Protopopoff et al.[50,51], where very high post-intervention

prevalence was reported towards the end of the trial, or in Accrombessi et al.[13,14], where the model underestimated the impact of pyrethroid-only ITNs at the beginning and at the end of the trial. Previous modelling attempts to reproduce these two trials with another malaria model faced similar issues and outcomes[36,40]. In Protopopoff et al.[50,51], some observations reported prevalence above 68% among children under 15 years old, reaching as high as 81% in one arm: it is likely that the occurrence of such high prevalence is due to external factors not included in the model. Mathematical models are simplified representations of reality, which is far more complex than can be perfectly predicted. It is therefore important to remember that, while such models can be useful at predicting general patterns and trends, they are not necessarily effective for forecasting precise future outcomes.

Third, our effectiveness cascades are quantified using the theoretical concept of vectorial capacity reduction. Although this metric does not directly reflect the burden of malaria, it has the advantage of being independent of the contextual factors that are not directly related to vector control. These factors include the acquired immunity in humans, case management and drug-based intervention coverage and climate variations. As the drivers of malaria transmission are non-linear, the same reduction in vectorial capacity can lead to different numbers of cases averted, depending on setting-specific factors (as can be seen from the variability in modelled effect sizes). Therefore, our cascades should not be interpreted as predictions of the public health impact of ITN products, but rather as a tool for identifying the relative importance of the factors responsible for the loss of impact in a given context. Moreover, due to the controlled nature of the RCT, some elements influencing ITN effectiveness are not included in our cascade, such as seasonal ITN use[62]. Nevertheless, provided that precise data on these quantities is available, the framework could be extended to include them.

Finally our model and associated tool do not incorporate considerations related to the management of insecticide resistance spread. Insecticide resistance is constantly evolving, even as a consequence of successful ITN interventions[63] and it can threaten the public health impact of ITNs. In our cascade, resistance is implicitly included in the 'entomological efficacy': this step reflects the properties of the ITNs in terms of killing and feeding inhibition under the resistance levels at the time and location where the EHTs were conducted. Building on other modelling approaches that include a relationship with bioassay mortality[41] or a dynamical representation of resistance spread[64] could help to extrapolate predictions more accurately to new geographies or the future. While this is beyond the scope of the current analysis, our model could be applied to more EHT data as they get generated, to follow how entomological efficacy estimates change in reaction to resistance spread.

In conclusion, we have quantified the loss of ITN impact along a cascade from entomological efficacy to population-level effectiveness, highlighting the importance of operational factors such as functional survival, ITN use or in-bed exposure. The performance of an ITN is dependent on several components beyond the insecticide that result in substantial differences in effectiveness. As these differences may be setting-specific due to different mosquito profiles, human behaviour, environment and net care behaviour, our framework allows the combination of multiple data sets to better understand essential components of product effectiveness and enables setting-specific selection of maximally effective ITN products for use in malaria control and elimination.

## Methods

### Estimating ITN entomological efficacy from experimental hut trial data

**Experimental hut trial data.** Data from eight EHT, conducted in Tanzania, Côte d'Ivoire and Benin, are used. The main characteristics of each included trial arm are presented in Table 1 and the reader is referred to the original publications for further details (or the Supplementary Tables 6.1 and 6.2 for unpublished trials). We only retained

the arms involving Interceptor® G2, Olyset® Plus and pyrethroid-only ITNs. Additionally, we only retained products for which measurements were available for both unwashed/new and washed/operationally-aged nets and we only kept the measurements with the highest number of washes or duration of field use.

In all trials, untreated nets are taken as the control. Each trial reports the number of mosquitoes found unfed-alive (UA), unfed-dead (UD), fed-alive (FA) and fed-dead (FD) immediately after collection and then held in a temperature-controlled facility to assess delayed mortality at 24 h (all trials except Nguessan et al.[48] and Sovegnon et al.[49]) and 72 h (all trials except BIT055 and Odufuwa et al.[46]). This information is summarised over all nights for each experiment in Supplementary Tables 2.1 and 2.2. The analyses are restricted to the dominant vector species collected (*Anopheles arabiensis* for Kibondo et al.[45], BIT080, Assenga et al.[44], BIT055 and Odufuwa et al.[46] and *Anopheles gambiae s.l.* for Assenga et al.[44], Nguessan et al.[48] and Sovegnon et al.[49]), due to low total numbers collected for other *Anopheles* species. In Martin et al.[47], the analysis was conducted on both *Anopheles gambiae s.l.* and *Anopheles funestus*. For each trial, the main analysis used the longest holding time to assess mortality (72 h for Kibondo et al.[45], BIT080 and Assenga et al.[44], Martin et al.[47], Nguessan et al.[48] and Sovegnon et al.[49]) and 24 h for BIT055 and Odufuwa et al.[46]. As a robustness check, the analysis using 24 h for all trials where the outcome was measured is presented in Supplementary Fig. 7.

**Statistical model.** The reduction in host availability (deterrence), the pre-prandial killing effect and the post-prandial killing effect are estimated using a Bayesian hierarchical model, adapted from Denz et al.[38] (see Fig. 5). In this model, each night $t$ and in each trial arm $i$, mosquitoes have the possibility to feed with rate $\alpha_i$ or die during host-seeking with rate $\mu_i$. Those who have fed have then a probability $p_{Ci}$ to survive after feeding. At the end of the night, mosquitoes are found in the four categories 'UA' ($UA_{i,t}$), 'UD' ($UD_{i,t}$), 'FA' ($FA_{i,t}$) and 'FD' ($FD_{i,t}$), with respective probabilities $(p_{UAi}, p_{UDi}, p_{FAi}, p_{FDi})$, with $p_{UAi} + p_{UDi} + p_{FAi} + p_{FDi} = 1$. The model can thus be written with the following equations (all notations are indicated in Table 2):

$$UA_{i,t}, UD_{i,t}, FA_{i,t}, FD_{i,t} \sim \text{Multinom}(p_{UAi}, p_{UDi}, p_{FAi}, p_{FDi}) \quad (1)$$

with

$$p_{UAi} = e^{-(\alpha_i + \mu_i)\tau} \quad (2)$$

$$p_{UDi} = \left(1 - e^{-(\alpha_i + \mu_i)\tau}\right) \frac{\mu_i}{\alpha_i + \mu_i} \quad (3)$$

$$p_{FAi} = \left(1 - e^{-(\alpha_i + \mu_i)\tau}\right) \frac{\alpha_i}{\alpha_i + \mu_i} p_{Ci} \quad (4)$$

$$p_{FDi} = \left(1 - e^{-(\alpha_i + \mu_i)\tau}\right) \frac{\alpha_i}{\alpha_i + \mu_i} (1 - p_{Ci}) \quad (5)$$

The effect of a given intervention is measured by comparing the rates $\alpha_i$ (feeding rate) and $\mu_i$ (mortality rate while host-seeking) and the survival probability after biting $p_{Ci}$ between the intervention arm (pyrethroid-only, chlorfenapyr or PBO net) and the control arm (untreated net, denoted with $i = 0$).

The parameter $\pi_i$ corresponds to the reduction in host availability for protected hosts in intervention arm $i$ and thus represents the deterrence effect. It is defined as follows:

$$\alpha_i = \alpha_0 (1 - \pi_i) \quad (6)$$

**Table 1 | Description of the experimental hut trials included in the analysis**

| Experimental hut trial | Location | Year | Mosquito species | Pyrethroid-only ITNs | Next-generation ITNs | Use status | Hut style |
|---|---|---|---|---|---|---|---|
| 1 Assenga et al.[44] | Tanzania (Ifakara, Morogoro) | Sept–Oct 2023 | An. arabiensis | MAGNet® (alpha-cypermethrin) | Interceptor® G2, Olyset® Plus | Unwashed and 20× washed | Ifakara, East African, West African and Rapley |
| 2 Kibondo et al.[45] | Tanzania (Ifakara, Morogoro) | Feb–Mar 2021 | An. arabiensis | Interceptor® (alpha-cypermethrin) | Interceptor® G2 | Unwashed and 20× washed | Ifakara |
| 3 BIT080 | Tanzania (Ifakara, Morogoro) | Apr–May 2023 | An. arabiensis | MiraNet® (alpha-cypermethrin) | Interceptor® G2 | Unwashed and 20× washed | Ifakara |
| 4 BIT055 | Tanzania (Ifakara, Morogoro) | Nov–Dec 2020 | An. arabiensis | Olyset® (permethrin) | Olyset® Plus | Unwashed and 20× washed | Ifakara |
| 5 Odufuwa et al.[46] | Tanzania (Ifakara, Morogoro) | Sept–Oct 2020 | An. arabiensis | PermaNet® 2.0 (deltamethrin) | Olyset® Plus | Unwashed and 20× washed | Ifakara |
| 6 Martin et al.[47] | Tanzania (Magu, Mwanza) | May 2020– Dec 2022 | An. gambiae s.l. | Interceptor® (alpha-cypermethrin) | Interceptor® G2Olyset® Plus | New and operationally-aged (after 36 months) | East African |
| 7 Martin et al.[47] | Tanzania (Magu, Mwanza) | May 2020– Dec 2022 | An. funestus | Interceptor® (alpha-cypermethrin) | Interceptor® G2Olyset® Plus | New and operationally-aged (after 36 months) | East African |
| 8 Assenga et al.[44] | Côte d'Ivoire | May–July 2023 | An. gambiae s.l. | MAGNet® (alpha-cypermethrin) | Interceptor® G2, Olyset® Plus | Unwashed and 20× washed | West African |
| 9 Nguessan et al.[48] | Benin (Cové, Zou) | Apr–July 2015 | An. gambiae s.l. | Interceptor® (alpha-cypermethrin) | Interceptor® G2 | Unwashed and 20× washed | West African |
| 10 Sovegnon et al.[49] (trial 1) | Benin (Za-Kpota, Zou) | Jul–Aug 2021 | An. gambiae s.l. | Interceptor® (alpha-cypermethrin) | Interceptor® G2 | New and operationally-aged (after 24 months) | West African |
| 11 Sovegnon et al.[49] (trial 2) | Benin (Za-Kpota, Zou) | Oct–Nov 2021 | An. gambiae s.l. | | Interceptor® G2 | New and operationally-aged (after 24 months) | West African |

When $\pi_i = 0$, the intervention has no deterrence effect and when $\pi_i = 1$, protected hosts escape mosquito bites entirely.

The parameter $\kappa_i$ corresponds to an increase in the killing rate while the mosquito is host-seeking in intervention arm $i$ and it is defined as follows:

$$\mu_i = \mu_0 + \alpha_0 * \kappa_i \tag{7}$$

When $\kappa_i = 0$, the intervention has no effect on host-seeking mortality and when $\kappa_i > 0$ mosquitoes have a higher mortality while host-seeking than in the control. A related indicator is the pre-prandial killing effect $\phi_i$, namely the reduction in the probability $p_{S_i}$ of surviving before biting (where $p_{S_i} = 1 - p_{UD_i}$). The pre-prandial killing effect can be obtained via the following formula (which can be sampled while fitting the Bayesian model):

$$\phi_i = 1 - \frac{p_{Si}}{p_{S0}} \tag{8}$$

This can be rewritten as

$$p_{Si} = p_{S0}(1 - \phi_i) \tag{9}$$

When $\phi_i = 0$, the intervention has no effect on the probability that mosquitoes survive and when $\phi_i = 1$, all mosquitoes die and none can feed the protected host. Because $p_{S_i}$ is a function of $\alpha_i$ and $\mu_i$, which themselves depend on $\pi_i$ and $\kappa_i$, the pre-prandial killing effect $\phi_i$ is itself a function of both $\pi_i$ and $\kappa_i$. It is required for subsequent mathematical modelling with OpenMalaria and will be used in the model to represent the potential of the intervention to kill mosquitoes before feeding.

Finally the parameter $\xi_i$ corresponds to the post-prandial killing effect in intervention arm $i$, whereby the probability of mosquito survival after biting is decreased by the intervention.

$$p_{C_i} = p_{C0}(1 - \xi_i) \tag{10}$$

When $\xi_i = 0$, the intervention has no effect on post-prandial survival and when $\xi_i = 1$ no mosquito survives after biting when in contact with the intervention.

The model is fitted with Stan (version 2.32.2) using the Rstan interface (https://cran.r-project.org/web/packages/rstan, version 2.32.7), using 3 MCMC chains with 6000 iterations each, including 3000 warm-ups. Prior distributions and definition ranges for each parameter are indicated in Table 2. The model is fitted independently for each EHT and for unwashed/new and 20 times washed/aged nets separately. Convergence diagnostic plots are presented in the Supplementary Figs. 8.1–8.10.

**Vectorial capacity reduction.** In the model by Chitnis et al.[65], which is the non-periodic analogue of the model used in OpenMalaria, the vectorial capacity can be calculated either in the absence or in the presence of ITNs in the population to quantify the reduction in vectorial capacity attributable to the ITN intervention[20]. A theoretical ideal vector control tool that would entirely remove the mosquito population corresponds to a 100% reduction in vectorial capacity, while a completely ineffective intervention corresponds to a 0% reduction. Vectorial capacity is computed using the AnophelesModel R package[20] (https://github.com/SwissTPH/AnophelesModel, version 1.1.0).

In order to account for the decay in ITN effectiveness over time, the model's equilibrium vectorial capacity is calculated for different time steps over a 3-year period. At each time step, effective coverage is updated to reflect functional survival and entomological efficacy is updated to reflect insecticidal durability. The reductions in vectorial capacity calculated for each time step are averaged to obtain a single summary value over the 3-year time frame.

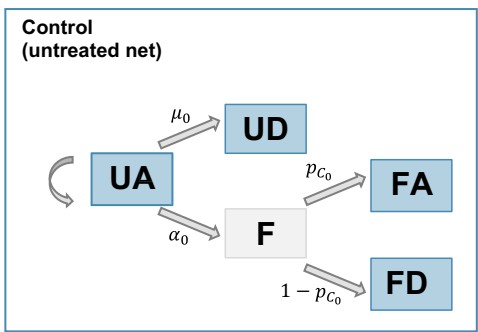
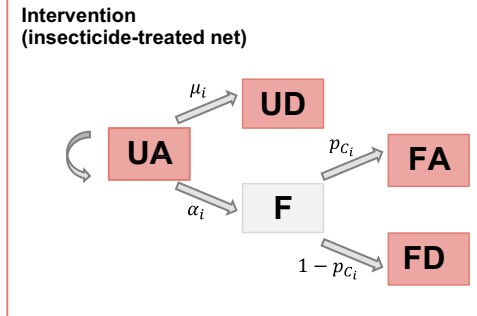

**Fig. 5 | Schematic representation of the model used for statistical estimation of entomological effects.** UA unfed alive, UD unfed dead, F fed, FD fed dead, FA fed alive. $\alpha$: feeding rate. $\mu$: death rate while host seeking. $p_c$: probability to survive after biting. For rates and probabilities, subscript 0 refers to the control arm and subscript $i$ refers to the intervention arm. The left panel (blue) represents the possible outcomes for a vector in presence of an unprotected individual in the control arm (untreated net). The right panel (red) represents the possible outcomes for a vector in presence of an individual protected by an ITN.

### Table 2 | Parameter definitions and associated prior distributions

| Quantity | Definition | Range/prior | UNIT |
|---|---|---|---|
| **Data** | | | |
| $UA_{i,t}$ | Number of mosquitoes unfed and alive at the end of night t in arm $i$ | ≥0 | mosquitoes |
| $UD_{i,t}$ | Number of mosquitoes unfed and dead at the end of night t in arm $i$ | ≥0 | mosquitoes |
| $FA_{i,t}$ | Number of mosquitoes fed and alive at the end of night t in arm $i$ | ≥0 | mosquitoes |
| $FD_{i,t}$ | Number of mosquitoes fed and dead at the end of night t in arm $i$ | ≥0 | mosquitoes |
| $F_{i,t}$ | Number of mosquitoes fed at the end of night t in arm $i$ ($FA_{i,t} + FD_{i,t}$) | ≥0 | mosquitoes |
| $\tau$ | Duration of host-seeking period for a night. The values of $\alpha_i$ and $\mu_i$ are scaled such that $\tau = 1$. | | day |
| **Fitted parameters** | | | |
| $\pi_i$ | Reduction in host availability ($\pi_0 = 0$) | Lognormal (0,5) on $1 - \pi_i$ | dimensionless |
| $\kappa_i$ | Increase in the killing rate while the mosquito is host-seeking ($\kappa_0 = 0$) | Lognormal (0,5) | dimensionless |
| $\xi_i$ | Post-prandial killing effect ($\xi_0 = 0$) | Uniform (0,1) | dimensionless |
| **Hyper parameters (fitted)** | | | |
| $a, m$ | Mean of $\log(\alpha_0\tau)$ and $\log(\mu_0\tau)$ | N (0,6) | dimensionless |
| $b$ | Mean of $\text{logit}(p_{C_0})$ | Logistic (0,1) | dimensionless |
| **Transformed quantities** | | | |
| $p_{UAi}$ | Probability to remain unfed and alive at the end of a night | [0,1] | dimensionless |
| $p_{FAi}$ | Probability to be fed and alive at the end of a night | [0,1] | dimensionless |
| $p_{FDi}$ | Probability to be fed and dead at the end of a night | [0,1] | dimensionless |
| $p_{UDi}$ | Probability to be unfed and alive at the end of a night | [0,1] | dimensionless |
| $p_{Si}$ | Probability to survive before biting | [0,1] | dimensionless |
| $p_{Ci}$ | Probability to survive after biting | [0,1] | dimensionless |
| $\alpha_i$ | Feeding rate | ≥0 | day⁻¹ |
| $\mu_i$ | Death rate while host seeking | ≥0 | day⁻¹ |
| $\phi_i$ | Pre-prandial killing effect | [0,1] | dimensionless |

Prior distributions are from Denz et al.[38].

In order to propagate the fitting uncertainty of the entomological efficacy estimates, vectorial capacity is calculated for 1000 random samples from the posterior distribution for each EHT. The posterior mean and the 95% credible intervals in the vectorial capacity estimates for each EHT were selected. Vectorial capacity computation is deterministic and the only source of uncertainty propagated is the posterior estimation of the reduction in host availability and the pre- and post-prandial killing effects.

In order to reflect entomological efficacy only (in Fig. 1), vectorial capacity is computed under ideal conditions, namely, ITN usage is assumed to be 100%, functional survival is assumed to be 3 years, insecticide decay is assumed to be absent and in-bed exposure is assumed to be 100%. Other conditions are explored subsequently in the cascade of effectiveness and the associated methodology is detailed below.

In order to formally assess the difference in entomological efficacy between products tested within the same EHT, we computed the difference between vectorial capacity estimates for 1000 parameter sets from the posterior distribution. We computed the 2.5 and 97.5 quantiles of the resulting distribution and assessed if this interval contained the value 0 or not (displayed in Supplementary Table 3).

**Reproducing in silico randomized control trial with a mathematical model of malaria transmission**
Four cluster RCTs comparing the efficacy of pyrethroid-only nets with next-generation nets including PBO and chlorfenapyr are considered

**Table 3 | Trial-specific assumptions for each of the four considered RCTs**

| Parameter | Assumptions and sources | Mosha et al.[11,12] | Protopopoff et al.[50,51] | Staedke et al.[52] | Accrombessi et al.[13,14] |
|---|---|---|---|---|---|
| Demography and age structure | As in OpenMalaria Utilities R package, version 23.2 | | | | |
| Mosquito species | Relative abundance from trial measurements, species-specific estimates for vector bionomics[20]. | An. funestus (95%)-An. gambiae (5%)[11] | An. arabiensis (4%)-An. funestus (4%)-An. gambiae (92%)[50] | An. arabiensis (5%)-An. funestus (25%)-An. gambiae (70%)[52] | An. gambiae s.l. 100% |
| In-bed exposure | Methodology by Golumbeanu et al.[20], averaging data from several studies (see Supplementary Fig. 11) | 69% | 77% | 76% | 77% |
| Seasonality profile | Based on rainfall data, extracted from CHIRPS via the R package chirps (https://cran.r-project.org/web/packages/chirps, v 0.1.4) using a 3 month lag between rainfall and EIR | Misungwi council (Mwanza), 2018–2022 | Muleba council (Kagera), 2014–2017 | Uganda, 2017–2019 | Zou department, 2019–2023 |
| Case management effective coverage | Calculated following Galactionova et al.[27] | 61.6% (2017 DHS survey, Mwanza) | 37.8% (2017 DHS survey, Kagera) | 67.1%[27] | 40.1% (2018, Zou[35]) |
| ITN usage pre-intervention | From trial measurements (and DHS survey for Accrombessi et al.[13,14]) | Py: 59.6%, IG2: 59.3%. OP: 61.7%[11] | Py: 30%. OP: 26%[50] | 39% in both arms | 80% in both arms (2018 DHS survey for Zou[67]) |
| ITN functional survival half-life | Inferred from the usage measured in the trial, using a Weibull function (see Supplementary material, section 10) | Py: 2.3 years. IG2: 2.4 years. OP: 1.7 years | Py: 2.7 years. OP: 2.4 years | Both arms: 4.2 years | Py: 4.6 years. IG2: 3.3 years |
| ITN usage post-intervention at deployment | Inferred from the measured usage in the trial, using a Weibull function (see Supplementary material, section 10) | Py: 77% IG2: 69% OP: 75% | Py: 77% OP: 80% | Both arms: 90% | Both arms: 100% |
| Entomological inoculation rate in the absence of interventions (EIR) | Calibrated to reproduce pre-intervention prevalence from RCTs, as detailed in the Supplementary material. Range of values across EHT assumptions for pyrethroid-only ITNS and calibration uncertainty | Py: 60–200 IG2: 46–196 OP: 46–196 | Py: 102–134 OP: 60–134 | Both arms: 14–20 | Py: 86–200 IG2: 46–200 |

'Py', 'IG2' and 'OP' refer to the pyrethroid-only arm, Interceptor® G2 arm and the Olyset® Plus arm respectively.

in this analysis: Protopopoff et al.[50,51] and Mosha et al.[11,12], conducted in Tanzania, Staedke et al.[52] conducted in Uganda and Accrombessi et al.[13,14] conducted in Benin. They are reproduced in silico using the OpenMalaria simulation platform (https://github.com/SwissTPH/openmalaria)[43]. This individual-based model accounts for the main specificities of malaria transmission dynamics in the human host (including immunity and superinfection) and the mosquito vector and it has been described extensively elsewhere[43].

The RCTs are modelled by adapting OpenMalaria to the setting-specific conditions of each trial location, using parameters as indicated in Table 3. The mathematical model's transmission intensity parameter entomological inoculation rate (EIR) is adjusted to represent the pre-intervention prevalence in each trial arm, following the methodology by Lemant et al.[35]. This approach relies on maximum likelihood to find the point estimate and profile likelihood methods to infer uncertainty ranges (see Supplementary material 9 for details). Resulting EIR estimates may vary across trial arms due to differing baseline prevalence and intervention coverages. RCT observations on post-intervention prevalence are used for validation. The simulations are conducted with OpenMalaria v44 and the OpenMalariaUtilities R package (https://github.com/SwissTPH/r-openMalariaUtilities, version 23.2), using a population of 10,000 individuals and 10 stochastic replicates for each scenario.

The agreement between model predictions and observations was assessed as in ref. 40 by computing a linear regression between the model predicted values ($X$) and the observed values ($Y$):

($Y_i = mX_i + 0$, where i are individual repeated measures for each trial) and reporting the coefficient m and the adjusted $R^2$. In the R software (as in ref. 40), the $R^2$ for that particular regression model is computed as $R^2 = 1 - \frac{\sum (y_i - \hat{y}_i)^2}{\sum y_i^2}$, based on the assumption that $E[Y] = 0$ when $X = 0$. This indicator was computed for prevalence measures, for effect sizes (relative difference between prevalence in the intervention arm and the pyrethroid-only control arm, i.e. $100 \times \frac{PR_0 - PR_i}{PR_0}$, where $PR_0$ is the prevalence in the control arm and $PR_i$ is the prevalence in the intervention arm) and for the relative change in prevalence compared to baseline (defined as $100 \times \frac{PR_{t=0} - PR_t}{PR_{t=0}}$, where $PR_{t=0}$ is the prevalence at baseline (pre-intervention) and $PR_t$ is the prevalence post-intervention).

**Entomological efficacy of ITNs in OpenMalaria.** Within OpenMalaria, the effectiveness of ITNs is calculated using an *Anopheles* life-cycle model[65,66]. In this model, the numbers of host-seeking, infected and infectious mosquitoes are simulated using deterministic difference equations that quantify their survival probability across five stages of the feeding cycle (host-seeking, feeding, searching for a resting place, resting and ovipositing), when they attack protected or unprotected human hosts.

ITNs are included in the model as a 'generic vector interventions' with the previously fitted values for $\pi$, $\phi$ and $\xi$ taken as deterrence, pre-prandial and post-prandial killing effects, respectively. For each EHT, the set of parameters with the highest posterior density is used. For RCTs conducted in East Africa (Mosha et al.[11,12], Protopopoff et al.[50,51] and Staedke et al.[52]), EHT estimates from Tanzania are used. For the RCT conducted in Benin (Accrombessi et al.[13,14]), EHT estimates from West Africa are used. Uncertainty in the effect size is propagated by using an ensemble of simulations based on parameters from the different EHTs. For each EHT, the parameter set with the highest posterior is used and we report the median and the envelope over all simulations across all EHT.

Therefore, the final uncertainty range includes (i) model stochasticity (10 seeds), (ii) uncertainty in initial transmission intensity resulting from profile likelihood methods[35] (see Supplementary material 9), (iii) variability across EHT datasets.

**Table 4 | Realistic estimates and perfect comparator used in the effectiveness cascades to represent the setting in Mosha et al. 2022; 2024)**

| | Realistic estimates to represent the setting in Mosha et al.[11,12] | Perfect comparator |
|---|---|---|
| Entomological efficacy | Deterrence, pre-prandial and post-prandial killing effects as fitted from the EHT data | Assuming a 100% reduction in vectorial capacity (no vectors) |
| Usage at distribution (Proportion of people sleeping under a net at night just after the net distribution) | 69% for Interceptor® G2 arm and 75% for Olyset® Plus (see Supplementary material, section 10) | 100% |
| Functional survival | Weibull decay function with half-life and shape parameters fitted to trial surveys on ITN use (see Supplementary material, section 10)<br>Half-life: 1.7 years for Olyset® Plus and 2.4 years for Interceptor® G2<br>Shape: 1.9 for Olyset® Plus and 2.4 for Interceptor® G2 | Weibull decay function with fixed half-life and shape parameters<br>Half-life: 3 years<br>Shape: 2 (assumed) |
| Insecticidal durability | Linear decay between estimates for unwashed and washed ITNs in the EHT, assuming that washed nets represent nets after $x$ years ($x = 3$ for all EHT but Sovegnon et al.[49] where $x = 2$). For EHT trial arm $i$ this corresponds to the following durations for deterrence, pre-prandial and post-prandial killing effects respectively:<br>$L_{\pi_i} = \dfrac{x}{1 - \pi_i^{washed}/\pi_i^{unwashed}}$<br>$L_{\phi_i} = \dfrac{x}{1 - \phi_i^{washed}/\phi_i^{unwashed}}$<br>$L_{\varepsilon_i} = \dfrac{x}{1 - \xi_i^{washed}/\xi_i^{unwashed}}$ | Estimates for unwashed nets are maintained throughout |
| Activity rhythms | Exposure in bed is 69% (see Table 3) | Exposure in bed is 100% |

**Epidemiological effectiveness of ITNs in OpenMalaria.** The Open-Malaria parameterisation includes various factors that influence ITN effectiveness beyond the sole entomological efficacy.

Firstly, the effectiveness of the ITNs depends on the proportion of individuals having access to and using them. In the model, the proportion of protected and unprotected hosts is informed by the proportion of individuals effectively using a net, inferred from RCT survey data on the proportion of individuals reporting using a net the night before the survey, as explained in the Supplementary material (section 10).

Secondly, the durability of ITNs over the 3-year period between distribution campaigns is accounted for via two mechanisms, namely insecticidal durability and functional survival. Insecticidal durability is informed by the entomological estimates for 20× washed (or aged) and unwashed nets. The estimates obtained for unwashed/new nets were used to represent the initial entomological efficacy. The estimates obtained for nets washed 20 times were used to represent the entomological efficacy after 3 years[8], assuming a linear decay over the 3-year time interval. The estimates obtained for operationally-aged nets were used to represent the entomological efficacy after 2 or 3 years, depending on the outcome measured in the trial, assuming a linear decay over the time interval. Functional survival is included by reducing the effective ITN usage over time. It is assumed that all individuals initially possessed study nets, which were then lost over time according to a decay function derived from the observed data. The decay distribution was assumed to be a Weibull function with half-life and shape parameter fitted to RCT data on ITN usage as explained in the Supplementary material (section 10), assuming half-life cannot exceed 3 years.

Thirdly, because insecticide treated nets can only protect hosts that are bitten while in bed, published data on activity rhythms for both humans and mosquitoes from Golumbeanu et al.[20] (see Supplementary Fig. 11) are used to quantify the exposure to mosquito bites while in bed (denoted $\varepsilon$). The effective usage of ITNs is thus multiplied by $\varepsilon$. In each case, all datasets originated from the country in which the RCT was conducted are averaged. If the database[20] does not contain data for the country, the average across all datasets originating from Africa is used instead.

## Cascades of ITN effectiveness

The cascade of effectiveness relies on the concept of vectorial capacity, defined and calculated as explained previously.

The effectiveness cascade includes 5 different factors in this order: entomological efficacy, functional survival, imperfect usage at distribution, insecticidal durability and activity rhythms. For each of these factors, we compute a realistic estimate (derived previously to reproduce the RCT by Mosha et al.[11,12]) and a perfect comparator corresponding to an ideal setting where there would be no loss of effectiveness due to the factor. For example, for ITN usage, realistic values from the trial are between 70 and 80%, while the perfect comparator would be 100%. All these estimates and their perfect comparator counterpart are detailed in Table 4.

To derive the effectiveness cascade, the vectorial capacity is first calculated considering the perfect comparator for each factor and then recalculated, replacing sequentially each factor by its realistic value.

In order to propagate the fitting uncertainty of the entomological efficacy estimates, vectorial capacity is calculated for 1000 random samples from the posterior distribution for each EHT. The posterior mean and the 95% credible intervals in the vectorial capacity across the samples of all EHT for each net type were selected (posterior mean and 95% credible interval for each EHT are presented in Supplementary Fig. 4).

The cascades presented in the main text are parameterised to represent the setting in Mosha et al.[11,12]), considering *Anopheles gambiae* vectors. Additionally, a dashboard is available to the reader to adapt the cascades to their own data.

Because there is no natural order between the different factors which all influence the model in non-linear ways, we conducted a sensitivity analysis by testing all possible ordering of the four factors (usage at distribution, functional survival, insecticidal durability and in-bed exposure). Under each permutation, the decrease in percentage points associated with each factor was computed, alongside the rank from 1 to 4 among the four factors.

## Ethics approval

Ethical approval from Ifakara Health Institute-Institutional Review Board and National Institute of Medical Research Tanzania were obtained, respectively IHI/IRB/No: 29-2020 and NIMR/HQ/R.8a/Vol.IX/3521 for BIT055 and IHI/IRB/No: 35-2021, NIMR/HQ/R.8a/Vol.IX/3957 for BIT080. For Assenga et al.[44], ethical approval was and IHI/IRB/No: 21–2023 and NIMR/HQ/R.8a/Vol.IX/4558 for Tanzania and Ref 081-22/MSHPCMU/CNESVS-km for Côte d'Ivoire. Volunteers were recruited on written informed consent. For all other studies, the analysis is based on previously published data.

**Reporting summary**

Further information on research design is available in the Nature Portfolio Reporting Summary linked to this article.

## Data availability

All formatted datasets are available together with the code at https://github.com/SwissTPH/ITNcascades/tree/main/EHT_fit/processed_data.

## Code availability

The transmission dynamics mathematical model is written in C++ and R and model code can be found here: https://github.com/SwissTPH/openmalaria (v44), here: https://github.com/SwissTPH/r-openMalariaUtilities (v 23.2) and here: https://github.com/SwissTPH/AnophelesModel (v1.1.0). Data manipulation code, input parameters and processing code in R are available here: https://github.com/SwissTPH/ITNcascades (also available on Zenodo here: https://doi.org/10.5281/zenodo.17396850).

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

## Acknowledgements

The authors want to thank Thomas Smith and Maximilian Gerhards for helpful discussion over the course of the project and Rémi Turquier for visualisation support. Calculations were performed at the sciCORE (http://scicore.unibas.ch/) scientific computing core facility at the University of Basel. This work was supported in whole or in part by the Gates Foundation [INV-030449, INV-068864 and INV-025569]. Grant numbers are INV-030449 (EP, CC, JL, MA, RG, MG), INV-068864 (EP, CC, JL, MA, RG, MG) and INV-025569 (NC). The conclusions and opinions expressed in this work are those of the author(s) alone and shall not be attributed to the Foundation. Under the grant conditions of the Foundation, a Creative Commons Attribution 4.0 License has already been assigned to the Author Accepted Manuscript version that might arise from this submission. The funders did not play any role in the study design, data collection and analysis, decision to publish, or preparation of the manuscript.

## Author contributions

This work was conceived by C.C., J.L., S.J.M. and E.P. Empirical data were produced by A.A., U.A.K., R.G.L., E.M., J.M., J.B.M., W.S.N., O.G.O., J.K.S., R.N. and S.J.M. C.C. performed the statistical analysis, with support from M.A. C.C. and J.L. performed the modelling simulations, with support from R.G. and M.G. C.C. performed the post-processing analysis, produced all figures and wrote the first draft. Expert guidance on analysis and interpretation were provided by S.J.M., N.C, A.R. and E.P. All authors contributed comments and approved the final version of the manuscript.

## Competing interests

A.A., U.A.K., R.G.L., E.M., J.M., J.B.M., W.S.N., O.G.O., J.K.S. and S.J.M. test vector control products for a range of manufacturers. The other authors declare no competing interests.
