## [Transparent Peer Review File · Nature Communications]

Cascades of effectiveness of next-generation insecticide-treated nets against malaria, from entomological trials to real-life conditions

Corresponding Author: Dr Clara Champagne

Version 0:

Reviewer comments:

Reviewer #1

(Remarks to the Author)

This work quantifies the various parameters that can contribute to the ultimate efficacy of insecticide treated nets for two classes of ITN in Tanzania and then provides a dashboard to consider these impacts elsewhere. This is a very useful addition to the malaria community and makes clear some of the different challenges we face when trying to understand the performance of different interventions.

Please see attached comments

(Remarks on code availability)

The code is nicely annotated and provides the scripts to enable a reader to recreate the analysis but the data are missing so it is currently not possible to recreate the analysis. Could the authors please share the data in the "formatted" ways or flag which of the shared files align with the code (perhaps simply add a data folder to the repository)

3 Data xlsx files are submitted with the manuscript but it is not clear what trials are shared

Reviewer #2

(Remarks to the Author)

This is a substantial piece of work addressing a topic of high importance in the malaria field. The approach is theoretically sound, and novel to the field. It's well-written, and I appreciate this work's close collaboration with primary data collection. However, due to methodological weaknesses, I cannot recommend it for publication at this time. I have offered detailed comments below with my concerns, in the hopes that they will be helpful for the authors as they consider future submissions of this work. I thank the authors for their time.

1. Calibration to pyrethroid-only nets

I understand that all your EHTs used pyrethroid-only nets as controls, and therefore it was only possible to calculate the impact of next-generation ITNs relative to pyrethroid-only nets in those studies, rather than estimating absolute impact for both types of net. It also makes sense how you use the relative differences found in the EHTs to supplement your pyrethroid-only fit in your OpenMalaria calibrations. What troubles me, however, is the absence of a rigorous calibration of pyrethroid-only papers in the model. I understand that model identifiability is a challenge, but your paper is *about* defining the ITN impact parameters of 1) reduction in host availability, 2) pre-prandial killing, and 3) post-prandial killing. The choice to fix two of those three parameters at zero for pyrethroid-only nets, with no justification, is not methodologically robust. The fact that this approach fails to fit in-sample pyrethroid-only net data in the Protopopoff study does not increase my confidence. In addition to my concern on a pure methodological level, setting these parameters to zero ensures that you attribute as much impact as possible to next-generation nets over pyrethroid-only nets, possibly overestimating their effectiveness. When assumptions such as this are made in the modeling space, they should generally be designed to make effect size estimates more conservative. To resolve this issue: have you searched the literature for EHTs in the area that use no nets, or untreated

nets, as a control, and the pyrethroid nets in your study as interventions? These may be older and require some adjustment for insecticide resistance, but they could assist with benchmarking.

I would also advise changing the language on line 322– I don't see how the current paper demonstrates anything about heterogeneity in pyrethroid-only nets. Rather, I would note the fact that you calibrate to a generic pyrethroid-only net as a limitation, given the known heterogeneity between net brands.

2. Cascade of effectiveness

I like the idea behind this section, and I appreciate the narrative framing, but several things concern me. First: if I understand correctly, you're claiming that a "perfect" net would induce 100% mortality in the entire mosquito population. Why make this choice? Nets were never designed to be so powerful, nor would it be realistic to expect them to be. I would advise that you either make this section about relative effectiveness declines from some arbitrary maximum, or use your modeling methodology to estimate a realistic maximum possible impact of a net, and show declining effectiveness from there.

Second, I was surprised that you only considered the ITN "effectiveness" cascade in a single order: first entomological efficacy, then functional survival, usage at distribution, etc. down to "actual" effectiveness. Unlike a cascade of medical care, in which there is a natural linear order to "leaks" in the system, the ITN efficacy cascade is unordered. You could just as easily have reduced In-bed exposure before you reduced entomological efficacy, and it would likely have had an impact on the results, because every step of the cascade is working off a smaller margin of potential effectiveness than the one before. At a minimum, I'd like to see this listed as a limitation– at a maximum, I'd like to see a sensitivity analysis of how the results change with different ordering.

Finally, I'm confused about why you focused only on vectorial capacity in the cascade, rather than putting this same thing in terms of malaria burden. In either case, I'd be interested to know what the actual vectorial capacity reduction and/or burden reduction was in these clinical trials, and how your results compare to that estimate. I'd recommend putting these actual data values in the main text/figures, not the supplement.

3. The Protopopoff Trial:

I respect you keeping the model fit to this trial in the paper as opposed to dropping it, but I'm confused about the way it's being framed. You're trying to pose it as a successful model fit that we should interpret, when unfortunately that's not what I'm seeing. I think you should leave it in, but the framing should change.

I've already discussed the fit to the pyrethroid-only nets above. Given the lack of agreement between data and in-sample calibration, I would advise toning back your language on lines 183-4 that the model predictions hit every data point but one. I'm also confused at why you treated the high value as an outlier for calibration– I understand that the data point is unusual, but do you have any evidence that it's not accurate? If not, it shouldn't be thrown out. In general, I would reframe the Protopopoff story as an important reminder that models do not perform equally well in all settings. Your model works beautifully against one trial, and not so well against another. If you can't fix that calibration, I would make an honest description of why that is, and encourage readers not to make too strong an interpretation of your Protopopoff model fits. This is an opportunity to be a good teacher about responsible model interpretation.

In addition to the above, I have a number of shorter points that I would recommend addressing before a future submission:

- Line 92: you use the term "EHT" before defining it.
- Intro: It wasn't until the discussion that I realized all these studies were from Tanzania– you barely mention it earlier in the paper. It would have been helpful to have more context in the intro about the Tanzanian setting for this work– where were the trials conducted, what's the environment etc
- The introduction had very helpful background information, but it said very little about your actual study until the very end. I'd advise using the introduction to prepare the reader for what they're about to see in the results, so they don't get overwhelmed with technical details.
- I'm afraid I found every entry in the textbox extremely confusing– can you rewrite these in your own words, with very clear definitions of how we should interpret each phrase in the context of this paper in particular?
- Throughout: I'm confused about the terms "entomological efficacy" and "vectorial capacity reduction". Are they the same? If so, please pick one and stick to it. If not, please clearly explain the difference.
- Fig 1: I'd recommend adding the greek letters to the subplot headers, to guide the readers when they read the methods. I'd also suggest adding more informative axis lines– such as one at 80%, given that you discuss this as a cutoff in the text.
- I'm leery to recommend the use of a p-value, but they might be helpful when you're describing how to interpret the mean differences + confidence intervals on lines 158-166.
- Lines 175-6: these lines state that entomological efficacy is a model input, but the methods section makes it sound like a model output– which is it?
- Figure 2: Why does the Protopopoff model fit have much larger confidence intervals, if it's calibrated to more data points than the Moshia trial?
- Line 351– please give a bit more detail about trial location, timing, etc
- Line 365: I would make clear in the limitations/discussion that these results are only for *An. arabiensis*, not only list it in the methods

- Line 398: what is kappa, and how does it relate to your other two killing params? Why is it there?
- Table 1: a diagram/decision tree would be helpful to understand how these parameters interact
- Line 434: How long is a time step?
- Table 2: why is the non-intervention counterfactual EIR different between trial arms?

(Remarks on code availability)

I did a very brief scan of the GitHub repo, and it looks well-formatted and tidy. I know this takes a lot of work to maintain, and it's much appreciated. I would recommend adding a folder with all the data shown in figures, and a script allowing users to replicate the plots precisely.

Reviewer #3

(Remarks to the Author)

This paper brings about an interesting solution for a very relevant problem: How to optimize the deployment of ITNs from an explicitly epidemiological perspective, considering virtually all relevant practical parameters associated with this important public-policy intervention. It combines the analysis of experimental hut trials (EHTs), which were validated against Randomized Clinical Trials (RCTs) and used for parameterization of a previously existing malaria-epidemiological modelling framework (OpenMalaria).

The article is generally a good read with a clear and concise message. There are a few tweaks regarding abbreviations that appear in the text without previous explanation. Even though some of them are obvious for those familiar with the field, it is important to make the text accessible to scientists outside the domain, as well. E.g., 'EHT' appears for the first time in the text on line 92; LLIN, line 98. I suggest a review of accessibility of abbreviations and further domain-specific terminology throughout the text (with the recognition that Textbox 1 covers a great deal of this issue already).

The method of EHT analysis, using a Bayesian framework is truly fascinating. It is a pity that the approach is not explicitly mentioned in the Results section, which leaves the reader skeptical about the validity of the analysis work. Highlighting Bayesian metrics in the Results section could be one way to avoid unnecessary confusion.

That being said, one of my main concerns is the limited number of trials included in the analysis and even smaller number brought into the model validation. I failed to find the right justification for the small number of trials in the text and, while I have suspicions on the reasons why, I would like to know that explicitly from the authors: Are there any other opportunities for an improved sampling of studies? It seems a bit of a stretch to base some of much of a potentially immensely useful set of recommendations on just two validations, one of which of very questionable fit (Protopopoff et al. 2018, 2023). To me, this is a major point that needs clear justification, consideration of impact on the results and associated recommendations, and a proposal of a way forward for a robust policy recommendation. There is one very strong claim on line 263, which – to me – needs substantial more backing on data.

A separate point, which does not pertain to the immediate epidemiological effect of ITN deployment, but is of potentially disastrous impact, is the role of resistance evolution of the vector population to the insecticides in consideration. This needs to be discussed and the work done in this area considered, as well. The strategies of ITN deployment are not just a short-term problem, and resistance can be a major driver of failure in the medium term. One particular aspect I would like to see discussed is the interplay between insecticide resistance and the cascade of effectiveness.

All in all, this paper proposes a tantalizing approach to improving the comparison and deployment of ITNs: A theoretical solution to a very practical and real problem. Yet, "extraordinary claims require extraordinary evidence". In order to recommend the acceptance of this paper, I need see the limited sample of studies addressed in the text. My recommendation is to include major revisions of the included studies to validate the very powerful claim this paper makes.

(Remarks on code availability)

Version 1:

Reviewer comments:

Reviewer #1

(Remarks to the Author)

Thanks for the revised draft and updated dashboard to review.

I think the work is improved for this revision. The inclusion of the additional EHTs is great and also the validation effort against additional studies increasing the potential applicability of the dashboard.

One question of interest: When simulating the trials for Tanzania (Moshia), did you match the replacement of ITNs throughout as was explored in Churcher et al 2024? This could have made a substantial impact and it would be interesting to hear how you addressed this (I could not see it in the supplement)

Only minor things:

line 182 - spelling degradation should be degradation

Please check supplementary info are referred to in order

Might be worth flagging the 25m point for the Protopopoff study that is missed in the pyrethroid only ITN arm.

(Remarks on code availability)

The github is well organised and flagged appropriately throughout the script and code repository. The processed data is a useful resource.

I have not run the full analysis due to time limitations - but it reads well for the spot checks I tried.

Reviewer #2

(Remarks to the Author)

I commend the authors for the remarkable work that went into revising this manuscript. I now believe it to be much stronger, and can recommend it for publication.

My only note is to ask for more detail on previous work in the field. On lines 83-90, the authors state that effectiveness cascades for vector control have been presented, but rarely formally quantified. In addition to a few site-specific, data-driven examples (refs 30-32), they cite one modeling paper that approaches this same problem (ref 33). However, they do not discuss how their modeling analysis is different from this previous study. I am aware of at least one other modeling analysis, from the 2022 World Malaria Report, which approaches a very similar question. I'd like the authors to please extend this paragraph to briefly discuss these two studies, and any other related modeling work in the field, and state clearly how their work expands upon or diverges from these prior analyses.

(Remarks on code availability)

The code repository continues to look tidy and well-organized. The authors state that they have included a folder with all the data shown in the figures, and a script allowing users to replicate the plots precisely, but I was not able to find it. Can the authors please label this directory more clearly, and directly reference it in the code availability section?

Reviewer #3

(Remarks to the Author)

Thank you for addressing, not only my concerns, but all of the reviewers' points. I am now satisfied with the quality of the work and the evidence backing its claims. My recommendation is for acceptance.

(Remarks on code availability)

REVIEWER COMMENTS

Reviewer #1 (Remarks to the Author):

This work quantifies the various parameters that can contribute to the ultimate efficacy of insecticide
treated nets for two classes of ITN in Tanzania and then provides a dashboard to consider these
impacts elsewhere. This is a very useful addition to the malaria community and makes clear some of
the different challenges we face when trying to understand the performance of different
interventions.

This is an excellent contribution. The draft is well written, presented and considered and highly
valuable to the malaria community. The work is novel in that it provides a flexible platform to
consider cascading loss of impacts from the delivery of ITNs of two classes that are in regular use
across malaria endemic regions. It is particularly useful to be able to consider the loss in impact due to
the product in contrast to that due to operational challenges such as ITN access, or usage and
exposure risks through human biting or mosquito activity patterns. I support its publication in Nature
Communications and believe it fits the journal well.

Many thanks for this very positive appreciation of our work.

I have only some minor suggestions / typo highlights to offer. But please consider the suggestions for
the app which might make it more translatable

Have you tested the process elsewhere where other EHTs used? It might be a worthwhile exercise to
confirm results are not substantially different? Appreciate that you would need to validate locally but
perhaps possible in Benin to at least provide 1 alternative species.

Many thanks for this remark. We have now expanded the work to include additional EHTs: three from
West Africa conducted in Benin (Nguessan et al. 2016, Sovegnon et al. 2024) and Côte d'Ivoire (Assenga
et al. 2025), and one from Tanzania (Martin et al. 2024) where outcomes can also be measured on both
*Anopheles gambiae* ss. and *Anopheles funestus*. We have also expanded the validation to include two
more RCTs, conducted in Benin (Accrombessi et al. 2023, 2024) and Uganda (Staedke et al. 2020).

Feedback on the app

At the moment, the first thing we see on the dashboard are the figures, which need some explanation
to interpret. Could you make the default page the "About" tab, and add a "Definitions" tab to this
page adjacent to the About tab. It would be excellent too to have an "Interpreting the outcome"
explainer to support users.

Thank you for these suggestions. The dashboard now includes a Description tab as initial page, with the
following text:

***"This dashboard provides effectiveness cascades for two insecticide-treated nets, namely Interceptor®
G2 and Olyset® Plus. Effectiveness is measured in terms of vectorial capacity reduction. The cascade
enables the user to quantify the loss of effectiveness attributable to five factors, namely
entomological efficacy, usage at distribution, functional survival, insecticidal durability and in-bed
exposure (as defined in Definition tab). The user can use the panel on the left-hand side to adapt the
assumptions and tailor the results to their setting of interest.***

**Interpreting the outcome: Using the default assumptions, the effectiveness of Interceptor® G2 nets**
**was above 90% when only entomological efficacy was considered. It dropped below 60% when all**
**other dimensions were included. In this example, the main factors responsible for the decline in**
**effectiveness were in-bed exposure (responsible for a drop of 13 points), insecticidal durability**
**(responsible for a drop of 12 points) and imperfect usage at distribution (responsible for a drop of 10**
**points).**

**A short summary of the assumptions and definitions is provided in the Assumption tab and all**
**methodological details are available in the following manuscript.”**

**The details for the various parameters are noted within the Assumption tab, I think it could be useful**
**to tell the user this. Under the Parameters title on the left of the screen, could you add; “For**
**definitions and assumptions on values please see Assumptions tab” or something similar.**

Thank you for this remark. We have added the following sentence above the left-hand side panel where
parameters can be edited:

***“For definitions and assumptions on parameter values please see Assumptions tab”***

**For attrition, it could be really useful to flag this tool: ITN Quantification that would allow countries**
**(outside of Tanzania) to gain information on their location.**

Thank you for sharing this useful resource. We have now mentioned it in the discussion:

***In practice, for other settings, online tools (e.g. [https://allianceformalariaprevention.com/itn-](https://allianceformalariaprevention.com/itn-quantification/)***
***quantification/)) are available to estimate functional survival and ITN needs over time using survey***
***data and stock-and-flow models (Bertozzi-Villa 2021, Koenker et al. 2023).***

And in the Assumptions tab, in the section on attrition:

***In practice, for other settings, online tools (e.g. ITN Quantification) are available to estimate***
***functional survival and ITN needs over time using survey data and stock-and-flow models (Bertozzi-***
***Villa et al. 2021 Koenker et al. 2023).***

**Could you have a button that automatically allows you to fill the same information for the two ITNs so**
**that the user can ensure no mistake in typing causes a comparison that is unmatched? I appreciate**
**there are measurable differences in the EHTs that would be retained, but might be useful for decisions**
**makers.**

Thank you for this suggestion. This functionality was added (now the user can make the Olyset® Plus
cascade use the same inputs as the ones chosen for Interceptor® G2 and vice versa).

**Linked to the only query above, could you add other experimental hut trial estimations from West**
**Africa? To the five described by BIT10348, BIT055, BIT08049, Odufuwa50, and Kibondo 51**
We have now included three EHTs from West Africa, conducted in Benin (Nguessan et al. 2016,
Sovegnon et al. 2024) and Côte d’Ivoire (Assenga et al. 2025).

**The Activity Patterns for the two ITNs are identical (which might be what you desire but flagging in**
**case as these are separate tabs)**

Thank you for this remark. We have now modified the app such that, if the patterns are identical for
Interceptor® G2 and Olyset® Plus, only one graphic is shown. If the two differ (because differing
assumptions have been chosen), two graphics are shown.

**Minor considerations**

**Introduction**

**Perhaps in the introduction, where you say: In 2023, overall access to ITNs was estimated at around**
**60% in Africa, with only five countries on the continent achieving the target of 80% of the population**
**sleeping under an ITN 4,18.**

**Add**

**In 2023, overall access to ITNs was estimated at around 60% in Africa, with only five of XX countries**
**with endemic malaria on the continent achieving the target of 80% of the population sleeping under**
**an ITN 4,18.**

**As this will give better context.**

Done.

**Perhaps in the introduction, where you say:**
**Various indicators exist to monitor insecticide resistance, ITN use, ITN durability and in-bed exposure**
**(see Textbox 1) but these indicators are measured independently of each other and therefore do not**
**provide information on the relative contribution of each factor to the overall public health impact of**
**ITNs.**

**Add**

**Various indicators exist to monitor insecticide resistance, ITN use, ITN durability and in-bed exposure**
**(see Textbox 1) but these indicators are usually measured independently of each other and therefore**
**do not provide information on the relative contribution of each factor to the overall public health**
**impact of ITNs.**

**As there are examples of studies that try to generate all these information, see xxxx.**

Done.

**Where you state**

**In particular, EHT data has been used to estimate the entomological efficacy of various vector control**
**tools and hence parameterize mathematical models of malaria 41–44.**

**It is probably worth adding two key publications in this space**

**Nash 2021**

**Challenger 2023**

Done

**Results**

**A typo to correct here:**

Entomological efficacy was quantified in four indicators: reduction in host availability (reflecting the
deterrent properties of the nets), pre-prandial killing effect (referring to mosquito mortality before
biting), post-prandial killing effect (referring to mosquito mortality after biting) and overall reduction
in vectorial capacity (representing to the potential for a given mosquito population to transmit
malaria by combining the previous three indicators).

Done

**Methods**

Where you mention “unfed-alive, unfed-dead, fed-alive and fed-dead” perhaps add the UA, UD, FA,
FD acronyms that align with the supplementary data files.

Done

There is a reference to Appendix 4 but I cannot find this document (potentially needs updating – I
think to Supplementary Table 1), similarly I think the ref to Appendix 2 should be Supplementary
Table 2? Think same throughout – presumably the journal instructions can advise.

Thank you for pointing this. We have standardized the references to the Supplementary material
throughout the manuscript.

**Typo: all notations are indicated inError! Reference source not found. (needs a space)**

Done

For the units, I think time is 1 day (or 1 night...) perhaps specify this if so in the Tables of parameters
time-1 where unit time is 1 day

Done

Just need to double check the order of Tables/Figs mentioned is as per article... I think Table 2 is
referenced after Table 3 – very minor point!!

Done

**Discussion**

The main findings are well presented and discussed, the only suggestions I have here is:
The only species included in the analysis is arabiensis – could you add a limitation to this effect. I
appreciate that were the methodology applied to other EHTs you could generate the entomological
efficacy reductions to build the framework, but at this stage, in the app particularly, the estimates are
from the 5 EHTs that consider arabiensis only.

Thank you for this suggestion. Our updated analysis now also includes one trial with results on *An.*
*funestus*, as well as three trials with *Anopheles gambiae s.l.* in West Africa. We have also added the
following limitation in the Discussion section:

***First, we estimated ITN entomological impact using EHTs from three countries: in Tanzania, results***
***were obtained for Anopheles arabiensis in five trials, and Anopheles funestus in one trial, in Benin and***

*Côte d'Ivoire, the results were obtained for Anopheles gambiae s.l. in three trials. Therefore, caution*
*should be exercised when extrapolating to other countries or vector species.*

**Congratulations on a great contribution.**

Many thanks again for your positive and constructive feedback.

**Reviewer #1 (Remarks on code availability):**

**The code is nicely annotated and provides the scripts to enable a reader to recreate the analysis but**
**the data are missing so it is currently not possible to recreate the analysis. Could the authors please**
**share the data in the "formatted" ways or flag which of the shared files align with the code (perhaps**
**simply add a data folder to the repository)**

We have now added a "processed_data" folder as part of the repository to enable users to replicate the
full analysis.

**3 Data xlsx files are submitted with the manuscript but it is not clear what trials are shared**

We have now included the data as part of the GitHub repository, and not as supplementary material
anymore, for increased readability.

**Reviewer #2 (Remarks to the Author):**

**This is a substantial piece of work addressing a topic of high importance in the malaria field. The**
**approach is theoretically sound, and novel to the field. It's well-written, and I appreciate this work's**
**close collaboration with primary data collection. However, due to methodological weaknesses, I**
**cannot recommend it for publication at this time. I have offered detailed comments below with my**
**concerns, in the hopes that they will be helpful for the authors as they consider future submissions of**
**this work. I thank the authors for their time.**

Thank you for highlighting the novelty and importance of our work and for providing constructive
suggestions for improvement. We have substantially revised the manuscript to address these remarks.
In particular, we have now estimated the entomological efficacy of pyrethroid-only ITNs directly from
EHT data, we have expanded the number of datasets used for fitting (four additional EHTs) and
validation (two additional RCTs) and we have provided a sensitivity analysis on the cascade approach.
These revisions are presented in details in the answers below.

**2. Calibration to pyrethroid-only nets**

**I understand that all your EHTs used pyrethroid-only nets as controls, and therefore it was only**
**possible to calculate the impact of next-generation ITNs relative to pyrethroid-only nets in those**
**studies, rather than estimating absolute impact for both types of net. It also makes sense how you use**
**the relative differences found in the EHTs to supplement your pyrethroid-only fit in your OpenMalaria**
**calibrations. What troubles me, however, is the absence of a rigorous calibration of pyrethroid-only**
**papers in the model. I understand that model identifiability is a challenge, but your paper is *about***
**defining the ITN impact parameters of 1) reduction in host availability, 2) pre-prandial killing, and 3)**
**post-prandial killing. The choice to fix two of those three parameters at zero for pyrethroid-only nets,**
**with no justification, is not methodologically robust. The fact that this approach fails to fit to in-**
**sample pyrethroid-only net data in the Protopopoff study does not increase my confidence. In**
**addition to my concern on a pure methodological level, setting these parameters to zero ensures that**
**you attribute as much impact as possible to next-generation nets over pyrethroid-only nets, possibly**
**overestimating their effectiveness. When assumptions such as this are made in the modeling space,**
**they should generally be designed to make effect size estimates more conservative. To resolve this**
**issue: have you searched the literature for EHTs in the area that use no nets, or untreated nets, as a**
**control, and the pyrethroid nets in your study as interventions? These may be older and require some**
**adjustment for insecticide resistance, but they could assist with benchmarking.**

Many thanks for this remark. In all the EHTs considered, there is an “untreated net” arm. We have
therefore re-fitted our model by taking the untreated net as control. This way, the pyrethroid-only nets
are now calibrated with the same approach and the same EHTs as the other nets.

All results have been updated to reflect these changes. In particular, the updated Figure 1 is now as
follows:

*Figure 1: Estimates of entomological efficacy for unwashed and washed/aged nets (Bayesian posterior mean and 95% credible*
 *intervals), compared to untreated nets as control. The horizontal lines represent the average across all EHT per net type for*
 *unwashed nets (solid line) and washed/aged nets (dashed lines). The various experimental huts considered are 1: Assenga et al.*
 *(Tanzania)⁴⁷, 2: Kibondo et al. (Tanzania)⁴⁸, 3: BIT080 (Tanzania), 4: BIT055 (Tanzania), 5: Odufuwa et al. (Tanzania)⁴⁹, 6: Martin*
 *et al. (An. gambiae, Tanzania)⁵⁰, 7: Martin et al., (An. funestus, Tanzania)⁵⁰, 8: Assenga et al. (Côte d'Ivoire)⁴⁷, 9: Nguessan et*
 *al. (Benin)⁵¹ 10: Sovegnon et al. (trial 1, Benin)⁵², 11: Sovegnon et al. (trial 2, Benin)⁵²*

**I would also advise changing the language on line 322– I don't see how the current paper**
 **demonstrates anything about heterogeneity in pyrethroid-only nets. Rather, I would note the fact**
 **that you calibrate to a generic pyrethroid-only net as a limitation, given the known heterogeneity**
 **between net brands.**

Thank you for pointing this. Because the efficacy of pyrethroid-only ITNs is now estimated from EHT
 data, we can comment on the variability of the obtained estimates, which is indeed large.

We have therefore revised the text in the discussion as follows:

*Substantial variation was observed across experimental hut trials, especially among the estimates for*
*pyrethroid-only ITNs. Some of the variability for pyrethroid-only ITNs may come from differences*
*between products (e.g. nets treated with permethrin, deltamethrin or alpha-cypermethrin that vary in*
*terms of their irritancy to mosquitoes), or differences in resistance levels across study sites. In*
*entomological trials the choice of active comparator is a critical determinant of relative ITN*
*performance⁵⁶. It is often assumed that all pyrethroids behave in the same way, but there is abundant*
*evidence that their mode of action varies from quite high irritancy to very low irritancy⁵⁷. Additionally,*
*EHT data often present large intrinsic variability in mosquito numbers and mortality^{9,42} that could*
*contribute to these variations. Our statistical model to infer entomological efficacy estimates from EHT*
*data is general and could be readily applied to other EHT datasets beyond the eight studies presented*
*here, to better understand the drivers of entomological efficacy and the variability between trials^{9,41,42}.*

We have also removed the sentences in the limitation that were referring to the way pyrethroid-only
ITNs are estimated, as this has now been modified.

**2. Cascade of effectiveness**

**I like the idea behind this section, and I appreciate the narrative framing, but several things concern**
**me. First: if I understand correctly, you're claiming that a "perfect" net would induce 100% mortality**
**in the entire mosquito population. Why make this choice? Nets were never designed to be so**
**powerful, nor would it be realistic to expect them to be. I would advise that you either make this**
**section about relative effectiveness declines from some arbitrary maximum, or use your modeling**
**methodology to estimate a realistic maximum possible impact of a net, and show declining**
**effectiveness from there.**

Thank you for raising this issue. In our cascade, the effectiveness is measured in terms of vectorial
capacity reduction. The vectorial capacity, defined as the total number of infectious mosquito bites
originating from each mosquito biting an infected human, quantifies the malaria transmission potential
of a given vector population. If vectorial capacity could be reduced by 100%, there would not be any
competent vectors. This is an ideal theoretical benchmark, which is unlikely to be achieved with any
single intervention. We have therefore re-labelled this maximum as "perfect vector control
(theoretical)" in our cascade. This also appears in the Methods section as follows:

***A theoretical ideal vector control tool that would entirely remove the mosquito population corresponds***
***to a 100% reduction in vectorial capacity, while a completely ineffective intervention corresponds to a 0%***
***reduction.***

Our estimate of entomological efficacy (first step in the cascade) reflects, in a way, the realistic
maximum possible impact that a given ITN could have based on its entomological properties and under
perfect operational conditions.

**Second, I was surprised that you only considered the ITN "effectiveness" cascade in a single order:**
**first entomological efficacy, then functional survival, usage at distribution, etc. down to "actual"**
**effectiveness. Unlike a cascade of medical care, in which there is a natural linear order to "leaks" in**
**the system, the ITN efficacy cascade is unordered. You could just as easily have reduced In-bed**

exposure before you reduced entomological efficacy, and it would likely have had an impact on the
 results, because every step of the cascade is working off a smaller margin of potential effectiveness
 than the one before. At a minimum, I'd like to see this listed as a limitation– at a maximum, I'd like to
 see a sensitivity analysis of how the results change with different ordering.

 Thank you for this remark. You are totally right that there is no natural linear order between the
 different factors, which all influence the model in non-linear ways. We have therefore conducted a
 sensitivity analysis on the cascade, testing all possible ordering of the four operational factors (usage at
 distribution, functional survival, insecticidal durability and in-bed exposure). The relative magnitude of
 the four factors remained unchanged, with only slight changes in the loss in percentage points. The
 results of this sensitivity analysis are now included in the Supplementary Tables 2.1 and 2.2:

	Point difference (min-max)	Rank (min-max)
Interceptor G2		
Insecticidal durability	3-5	3-4
In-bed exposure	12-15	1-2
Usage at distribution	12-15	1-2
Functional survival	4-5	3-4
Olyset Plus		
Insecticidal durability	2-3	4-4
In-bed exposure	14-17	1-2
Usage at distribution	11-13	3-3
Functional survival	15-20	1-2

*Table 2.1. Sensitivity analysis on the ordering of the cascade, taking the average across the EHT conducted in Tanzania. Only the*
 *posterior maximum for EHT was used. Reading note: Across all permutations of the cascade items, the point difference*
 *associated with "Insecticide durability" was between 3 and 5 points for Interceptor G2. This factor was ranked as 3rd or 4th most*
 *important among the 4 factors considered.*

	Interceptor G2		Olyset Plus	
	Point difference (min-max)	Rank (min-max)	Point difference (min-max)	Rank (min-max)
Assenga et al.				
Insecticidal durability	0-1	4-4	-1--1	4-4
In-bed exposure	13-16	1-2	14-18	1-2
Usage at distribution	13-16	1-2	11-14	3-3
Functional survival	4-5	3-3	16-21	1-2
Kibondo et al.				
Insecticidal durability	0-1	4-4		
In-bed exposure	13-16	1-2		
Usage at distribution	13-16	1-2		
Functional survival	4-5	3-3		
Odufuwa et al.				
Insecticidal durability	-2-0	4-4		
In-bed exposure	12-15	1-2		
Usage at distribution	12-15	1-2		
Functional survival	4-5	3-3		
BIT055				
Insecticidal durability			0-1	4-4
In-bed exposure			14-18	1-2
Usage at distribution			11-14	3-3
Functional survival			16-21	1-2
BIT059				
Insecticidal durability			1-1	4-4
In-bed exposure			14-18	1-2
Usage at distribution			11-14	3-3
Functional survival			16-20	1-2
Martin et al. (An. funestus)				
Insecticidal durability	8-11	2-3	6-9	4-4
In-bed exposure	10-15	1-3	12-15	1-2
Usage at distribution	10-15	1-3	10-12	3-3
Functional survival	4-5	4-4	13-18	1-2
Martin et al. (An. gambiae)				
Insecticidal durability	8-13	1-3	4-6	4-4
In-bed exposure	9-14	1-3	13-16	1-2
Usage at distribution	9-14	1-3	9-13	3-3
Functional survival	3-5	4-4	15-19	1-2

Table 2.2. Sensitivity analysis on the ordering of the cascade, for each EHT conducted in Tanzania. Only the posterior maximum
for EHT was used. Reading note: Across all permutations of the cascade items, the point difference associated with "Insecticide
durability" was between 0 and 1 points for Interceptor G2 in the data from Assenga et al. 2025 (Tanzania). This factor was
always ranked 4th most important among the 4 factors considered.

This is commented in the results section:

***These results are robust to changes in the ordering of the cascade: the relative importance of the four***
***factors remained broadly similar, with changes in the loss in percentage points in the same order of***
***magnitude as the Bayesian credible intervals from Error! Reference source not found. and the ordering***
***of the factors remaining generally the same (See Supplementary tables 2.1 and 2.2).***

And the methods section:

***Because there is no natural order between the different factors which all influence the model in non-***
***linear ways, we conducted a sensitivity analysis by testing all possible ordering of the four factors***
***(usage at distribution, functional survival, insecticidal durability and in-bed exposure). Under each***
***permutation, the decrease in percentage points associated with each factor was computed, alongside***
***the rank from 1 to 4 among the four factors.***

**Finally, I'm confused about why you focused only on vectorial capacity in the cascade, rather than**
**putting this same thing in terms of malaria burden. In either case, I'd be interested to know what the**
**actual vectorial capacity reduction and/or burden reduction was in these clinical trials, and how your**
**results compare to that estimate. I'd recommend putting these actual data values in the main**
**text/figures, not the supplement.**

Thank you for this suggestion. All considered trials reported prevalence as an outcome (primary
outcome in 3 out of 4 trials), therefore this is the indicator we used to assess the impact of the
interventions on malaria. We computed the effect sizes (reduction in malaria prevalence in the
intervention arm compared to the control arm) both in the RCT data and in our model estimates and
compared them in a scatter plot. We also computed the relative change compared to baseline (change
in malaria prevalence after intervention compared to the pre-intervention level). This graphic is now
included in the main text of the article.

*Figure 2. Comparison of observed values in RCTs (y-axis) and modelled estimates (x-axis), considering validation points only. The*
 *black dashed line indicates the Y=X line, i.e. perfect agreement. A. Observed and modelled prevalence, with associated*
 *uncertainty intervals. On the x-axis, error lines represent model uncertainty associated with EHT fitting, transmission intensity*
 *calibration and model stochasticity (see Methods section for methodological details). On the y-axis, error lines represent 2*
 *empirical standard deviations from the point estimate. B. Observed and modelled effect sizes (relying on the median model*

prediction across EHT trials, transmission intensity calibration and model stochasticity). Effect size is defined as $100 * \frac{PR_0 - PR_i}{PR_0}$,
where PR_0 is the prevalence in the control arm and PR_i is the prevalence in the intervention arm. C. Observed and modelled
relative change compared to baseline (relying on the median model prediction across EHT trials, transmission intensity
calibration and model stochasticity). Relative change compared to baseline is defined as $100 * \frac{PR_{t=0} - PR_t}{PR_{t=0}}$, where $PR_{t=0}$ is the
prevalence at baseline (pre-intervention) and PR_t is the prevalence post-intervention.

And it is associated with the following text:

***“Nonetheless, there was a general agreement between observations and model estimates (see Figure***
***2): in the linear regression between observed and modelled values, the adjusted R² was 0.94 for***
***prevalence, 0.64 for effect sizes and 0.56 for relative comparison to pre-intervention levels. The***
***regression coefficients for each indicator were 0.98, 1.18 and 0.80, respectively.”***

Both observed and modelled effect sizes display substantial variability within and across trials. This can
be driven by a multiplicity of factors, including pre-intervention transmission levels, intervention
coverages, non-linearities in malaria transmission and setting-specific differences in environmental and
socio-economic factors. Therefore, these metrics are not suitable to explore more generally the drivers
of ITN impact, as we wish to do with our effectiveness cascade.

Vectorial capacity reduction, however, is a theoretical quantity related to the basic reproduction
number R_0 , but which focuses on the aspect related to the vector side of the parasite’s transmission
cycle. This way, it relies only on parameters related to vectorial transmission and vector control
interventions (e.g. killing, repellency, durability of vector control products, vector control coverage) and
it enables comparisons independently of other contextual factors not directly related to vector control
(underlying EIR, case management coverage, etc.).

For the same reasons that RCTs don’t report measures of R_0 , it is not possible to measure vectorial
capacity directly in the field, and therefore compare our estimates with those from the RCTs. This
outcome is an extrapolation from our model, that we think is a useful tool for decision-makers.

We have added a limitation in the discussion to highlight the fact that vectorial capacity is only a
theoretical metric and not directly transposable into burden estimates:

***Third, our effectiveness cascades are quantified using the theoretical concept of vectorial capacity***
***reduction. Although this metric does not directly reflect the burden of malaria, it has the advantage of***
***being independent of the contextual factors that are not directly related to vector control. These***
***factors include the acquired immunity in humans, case management and drug-based intervention***
***coverage, and climate variations. As the drivers of malaria transmission are non-linear, the same***
***reduction in vectorial capacity can lead to different numbers of cases averted, depending on setting-***
***specific factors (as can be seen from the variability in modelled effect sizes). Therefore, our cascades***
***should not be interpreted as predictions of the public health impact of ITN products, but rather as a***
***tool for identifying the relative importance of the factors responsible for the loss of impact in a given***
***context.***

**3. The Protopopoff Trial:**

**I respect you keeping the model fit to this trial in the paper as opposed to dropping it, but I’m**
**confused about the way it’s being framed. You’re trying to pose it as a successful model fit that we**
**should interpret, when unfortunately that’s not what I’m seeing. I think you should leave it in, but the**
**framing should change.**

Thank you for this comment. In the revised manuscript, we have kept the Protopopoff example, but we have also expanded the validation analysis to include the two remaining published RCTs, conducted in Benin and Uganda, therefore increasing the examples on which the model is tested.

I've already discussed the fit to the pyrethroid-only nets above. Given the lack of agreement between data and in-sample calibration, I would advise toning back your language on lines 183-4 that the model predictions hit every data point but one. I'm also confused at why you treated the high value as an outlier for calibration– I understand that the data point is unusual, but do you have any evidence that it's not accurate? If not, it shouldn't be thrown out.

Thank you for this remark. Following your above suggestion, the model has been updated to infer the impact of pyrethroid-only ITNs from EHT data as well. With this change, all post-intervention prevalence points are treated equally as validation points (and no point is excluded as an outlier anymore). Additionally, there was some improvement in the results for pyrethroid-only ITNs in the Protopopoff trial, and now only three observations post-intervention are missed in November 2016, June 2017 and November 2017. The observation in June 2017 is also missed in the Olyset Plus arm (as before). It is important to note, though, that these three prevalence measurements are high, and above the pre-intervention prevalence levels.

Therefore, Figure 2 is now as follows:

We have also revised the text in the results section to better reflect the limitations of our validation.

Over all trials, the uncertainty intervals for the model predictions and observations overlapped at all time points for Interceptor® G2, and all time points but two for Olyset® Plus. In the trial by Staedke et al.⁵⁵, the second observation was slightly missed. In the trial by Protopopoff et al.^{53,54}, the observation at 28 months in the intervention arm was largely missed; however, the observed prevalence at that date was abnormally high in both the control and intervention arm, the estimates being higher than all other observations including the pre-intervention estimate. For pyrethroid-only ITNs, although the models did reproduce general trends, seven out of 17 observed prevalence points were not captured by the model estimates. In Mosh et al.^{11,12}, two observations were slightly missed. In Protopopoff et al.^{53,54}, three

observations with high prevalence toward the end of the trial were missed. In Accrombessi et al.^{13,14},
the impact of pyrethroid-only ITNs was underestimated at the beginning and at the end of the post-
intervention period.

*Over all trials, the uncertainty intervals for the model predictions and observations overlapped at all*
*time points for Interceptor G2, and all time points but two for Olyset Plus. In the trial by Staedke et*
*al.⁵⁶, the second observation was slightly missed. In the trial by Protopopoff et al.^{54,55}, the observation*
*at 28 months in the intervention arm was largely missed; however, the observed prevalence at that date*
*was abnormally high in both the control and intervention arm, the estimates being higher than all other*
*observations including the pre-intervention estimate. For pyrethroid-only ITNs, although the models did*
*reproduce general trends, seven out of 17 observed prevalence points were not captured by the model*
*estimates. In Moshia et al.^{12,13}, two observations were slightly missed. In Protopopoff et al.^{54,55}, three*
*observations with high prevalence toward the end of the trial were missed. In Accrombessi et al.^{14,15},*
*the impact of pyrethroid-only ITNs was underestimated at the beginning and at the end of the post-*
*intervention period.*

**In general, I would reframe the Protopopoff story as an important reminder that models do not**
**perform equally well in all settings. Your model works beautifully against one trial, and not so well**
**against another. If you can't fix that calibration, I would make an honest description of why that is,**
**and encourage readers not to make too strong an interpretation of your Protopopoff model fits. This**
**is an opportunity to be a good teacher about responsible model interpretation.**

Thank you for this feedback. We have revised the wording of our discussion sections to better nuance
our claims as follows:

*At the population level, we validated our dynamical model's ability to reproduce the reduction in*
*malaria prevalence observed in four RCTs conducted in Tanzania, Uganda, and Benin. These trials*
*consistently showed greater effectiveness of next-generation ITNs compared to pyrethroid-only nets,*
*though the duration of protection varied substantially across net types and settings. Despite some*
*discrepancies, our model reproduced the overall magnitude of the impact of next-generation ITNs on*
*prevalence and captured much of the associated uncertainty. When considering the effect sizes*
*between next-generation and pyrethroid-only ITNs and the relative comparison between pre- and*
*post-intervention levels, substantial variability was observed in both the observed and modelled*
*values within and across trials. Although the model did not capture perfectly the observed*
*heterogeneity for these two indicators, there was agreement in the general trends. Overall, this*
*evidence from RCTs is in line with the results observed the effectiveness pilots conducted in Burkina Faso,*
*Mozambique, and Rwanda, that also observed a superior effectiveness of chlorfenapyr and PBO nets*
*relative to pyrethroid ones for at least two years after distribution¹⁶. Less clear results were observed in*
*Nigeria where diverging trends in prevalence and incidence could be attributed to data quality issues,*
*confounding interventions and external factors, and where ITNs usage was low¹⁶. Therefore, our*
*updated parameterization of the OpenMalaria model, which has been tested against RCT data and*
*demonstrates promising alignment especially for prevalence outcomes, can now be used to predict*
*potential ITN impact in places where there have not been RCTs. It can also be used for subnational*
*tailoring of interventions and/or country-wide predictions of the impact of malaria interventions^{35,58}.*

And added this paragraph in the limitations section of the discussion:

***Second, despite an overall good agreement between observations and model predictions, there were***
***divergences in some instances, notably in Protopopoff et al.^{54,55}, where very high post-intervention***
***prevalence was reported towards the end of the trial, or in Accrombessi et al.^{14,15}, where the model***
***underestimated the impact of pyrethroid-only ITNs at the beginning and at the end of the trial.***
***Previous modelling attempts to reproduce these two trials with another malaria model faced similar***
***issues and outcomes^{38,42}. In Protopopoff et al.^{54,55}, some observations reported prevalence above 68%***
***among children under 15 years old, reaching as high as 81% in one arm: it is likely that the occurrence***
***of such high prevalence is due to external factors not included in the model. Mathematical models are***
***simplified representations of reality, which is far more complex than can be perfectly predicted. It is***
***therefore important to remember that, while such models can be useful at predicting general patterns***
***and trends, they are not necessarily effective for forecasting precise future outcomes.***

**In addition to the above, I have a number of shorter points that I would recommend addressing**
**before a future submission:**

**- Line 92: you use the term “EHT” before defining it.**

Corrected

**- Intro: It wasn't until the discussion that I realized all these studies were from Tanzania– you barely**
**mention it earlier in the paper. It would have been helpful to have more context in the intro about the**
**Tanzanian setting for this work– where were the trials conducted, what's the environment etc**

Thank you for this remark. We have now included three additional EHTs conducted in Benin and Côte
d'Ivoire, as well as the two remaining published RCTs on new-generation ITNs, conducted in Uganda
(Staedke et al. 2020) and Benin (Accrombessi et al. 2023, 2024). The work is therefore not restricted to
Tanzania anymore.

**- The introduction had very helpful background information, but it said very little about your actual**
**study until the very end. I'd advise using the introduction to prepare the reader for what they're**
**about to see in the results, so they don't get overwhelmed with technical details.**

Thank you for this remark. We have reformulated the final paragraph of the introduction to give a more
precise description of the analyses conducted:

***In this work, we quantify the epidemiological impact of two next-generation ITNs, namely Olyset® Plus***
***(PBO-pyrethroid) and Interceptor® G2 (chlorfenapyr-pyrethroid). First, the entomological efficacy of the***
***ITNs is estimated using a Bayesian statistical model fitted to data from eight EHT conducted in***
***Tanzania, Côte d'Ivoire and Benin. Secondly, these entomological efficacy estimates are incorporated***
***into an established mathematical model of malaria transmission (the OpenMalaria simulation***
***platform⁴⁵) and the model's capacity to predict population-level prevalence is validated against the***
***outcomes of four RCTs conducted in Tanzania, Benin and Uganda. Finally, we use the mathematical***
***model to quantify the impact of both net types along the different steps of the effectiveness cascade and***
***identify the relative contributions of the main factors responsible for the loss of effectiveness.***

- I'm afraid I found every entry in the textbox extremely confusing– can you rewrite these in your own
words, with very clear definitions of how we should interpret each phrase in the context of this paper
in particular?

Thank you for this remark. We have reformulated the definitions and added two entries on “ITN
effectiveness” and “Vectorial capacity” to increase clarity. The revised textbox is as follows:

**Entomological Efficacy:** *capacity of a given vector control product to reduce mosquito transmission*
*potential, through the alteration of mosquito biting, mortality and/or fecundity, under ideal operational*
*conditions. It can be quantified as the relative reduction in vectorial capacity through deploying the*
*vector control product under ideal operational conditions.*

**ITN effectiveness:** *capacity of a given ITN product to reduce mosquito transmission potential under user*
*conditions. It differs from entomological efficacy in that insecticidal durability, functional survival, ITN*
*use and in-bed exposure are also accounted for. It can be quantified as the relative reduction in vectorial*
*capacity through deploying the ITN under realistic operational conditions.*

**Vectorial capacity:** *Total number of potentially infectious bites originating from all the mosquitoes biting*
*a single perfectly infectious (i.e. all mosquito bites result in infection) human on a single day. It*
*represents the potential for a given mosquito population to transmit malaria.*

**ITN use:** *Proportion of the population sleeping under a net the previous night. It is the result of both*
*“access” to a net (proportion of the population with access to an ITN for every two persons that slept in*
*the household the previous night) and “use given ownership” (proportion of the population with ITNs*
*that slept under a net the previous night).*

**Insecticidal durability:** *“Retention of insecticidal efficacy of the ITN”⁴⁶.*

**Functional survival:** *“An estimate of the physical lifespan of an ITN product in the field [...] It is measured*
*as an ITN that is present in use and in serviceable condition”⁴⁶. It is the result of the ITN's physical*
*integrity declining due to damage and attrition, that results in nets being discarded when the user no*
*longer regards them as useful.*

**In-bed exposure:** *The proportion of vector bites occurring indoors during sleeping hours out of all*
*potential bites in day (indoors and outdoors), for an unprotected individual. It represents “the maximum*
*possible personal protection any intervention targeting sleeping spaces could provide”⁴⁶.*

- Throughout: I'm confused about the terms “entomological efficacy” and “vectorial capacity
reduction”. Are they the same? If so, please pick one and stick to it. If not, please clearly explain the
difference.

Thank you for raising this point. In our analysis, “entomological efficacy” refers to the capacity of the ITN
to reduce mosquito transmission potential under ideal operational conditions. It is to be contrasted with
“ITN effectiveness”, which also includes in operational factors such as insecticidal durability, functional

survival, ITN use and in-bed exposure. Vectorial capacity reduction is the metric used to quantify both
effects in our mathematical model.

For better consistency across sections, we have renamed the panel in Figure 1 to “entomological
efficacy”.

Additionally, as mentioned above, the updated textbox now defines precisely these three terms as
follows:

**Entomological Efficacy:** capacity of a given vector control product to reduce mosquito transmission
potential, through the alteration of mosquito biting, mortality and/or fecundity, under ideal operational
conditions. It can be quantified as the relative reduction in vectorial capacity through deploying the
vector control product under ideal operational conditions.

**ITN effectiveness:** capacity of a given ITN product to reduce mosquito transmission potential under user
conditions. It differs from entomological efficacy in that insecticidal durability, functional survival, ITN
use and in-bed exposure are also accounted for. It can be quantified as the relative reduction in vectorial
capacity through deploying the ITN under realistic operational conditions.

**Vectorial capacity:** Total number of potentially infectious bites originating from all the mosquitoes biting
a single perfectly infectious (i.e. all mosquito bites result in infection) human on a single day. It
represents the potential for a given mosquito population to transmit malaria

- Fig 1: I'd recommend adding the greek letters to the subplot headers, to guide the readers when they
read the methods. I'd also suggest adding more informative axis lines– such as one at 80%, given that
you discuss this as a cutoff in the text.

We have now added the Greek letters to the subplot headers and the y-axis now includes more labels
(20%, 40%, 60% and 80%).

- I'm leery to recommend the use of a p-value, but they might be helpful when you're describing how
to interpret the mean differences + confidence intervals on lines 158-166.

Thank you for this comment. Because our statistical model relies on the Bayesian framework, instead of
providing p-values, we computed the difference in vectorial capacity reduction between the 2 products
over 1000 parameter sets from our posterior distribution. We assessed whether the value 0 was
between the 2.5% and 97.5% quantiles of that distribution to assess difference in impact. The table
containing these intervals is included in Supplementary material and reads as follows:

EHT	Difference in vectorial capacity reduction (95% credible interval)		
	Interceptor G2 - Pyrethroid	Olyset Plus - Pyrethroid	Interceptor G2 – Olyset Plus
1 Assenga et al. ³ (Tanzania)	-0.8% – 2.5 %	-1.5% – 1.7 %	-1.15% – 2.71 %
Kibondo et al. ⁴	1.6% – 7.8 %
BIT080	3.3% – 8.5 %
BIT055		0.7% – 3.0 %
Odufuwa et al. ⁶		8.1% – 15.3 %

Martin et al. ⁷ (An. gambiae)	13.1% – 59.4 %	12.2% – 58.9 %	-4.3% – 5.0 %
Martin et al. ⁷ (An. funestus)	5.6% – 55.4 %	-2.9% – 50.7 %	-0.7% – 17.0 %
Assenga et al. ³ (Côte d'Ivoire)	7.6% – 12.6 %	2.5% – 7.7 %	4.0% – 6.1 %
Nguessan et al. ⁸	10.6% – 16.6 %

*Supplementary Table 3.2. Difference between vectorial capacity estimates for 1000 parameter sets from the posterior*
*distribution (2.5 and 97.5 quantiles of the distribution), to assess the difference in entomological efficacy between ITNs*
*evaluated within the same EHT. Only unwashed/new ITNs are considered. Negative values are highlighted in red: these indicate*
*that the interval contains the value 0 and therefore that we cannot conclude to a difference in entomological efficacy between*
*the two ITNs given statistical uncertainty.*

It is commented in the Results section as follows:

And comm
***In all studies but two (Assenga et al.⁴⁸, Tanzania data and Martin et al.⁵¹ for *An. funestus* mosquitoes),***
***the difference in vectorial capacity reduction between next-generation ITNs and pyrethroid-only ITNs***
***was positive in more than 95% of the simulations, indicating the superiority of next-generation ITNs in***
***terms of entomological efficacy (see Supplementary table 3.2). In the studies which included both***
***types of next-generation net within the same trial in Tanzania (Assenga et al.⁴⁸ and Martin et al.⁵¹),***
***the difference in vectorial capacity reduction estimates between Interceptor® G2 and Olyset® Plus had***
***a 95% credible interval containing the value 0. This indicates that, considering the statistical***
***uncertainty, we could not conclude that there was a difference in impact between these two products.***
***In the EHT conducted in Côte d'Ivoire⁴⁸, however, the interval did not contain 0 and a higher***
***entomological efficacy was measured for Interceptor G2.***

The approach is also explained in the Methods section:

***In order to formally assess the difference in entomological efficacy between products tested within the***
***same EHT, we computed the difference between vectorial capacity estimates for 1000 parameter sets***
***from the posterior distribution. We computed the 2.5 and 97.5 quantiles of the resulting distribution***
***and assessed if this interval contained the value 0 or not (displayed in Supplementary table 3.2).***

- Lines 175-6: these lines state that entomological efficacy is a model input, but the methods section
males it sound like a model output– which is it?

Thank you for this remark. The entomological efficacy estimates are an output of section 1 (statistical
model) and they are used as an input in the following section (mathematical model). We modified the
phrasing to reduce the confusion:

***The aforementioned estimates (outputs displayed in Figure 1), alongside setting-specific information***
***related to epidemiological conditions and ITN effectiveness, were then included in the OpenMalaria***
***mathematical model of malaria transmission.***

- Figure 2: Why does the Protopopoff model fit have much larger confidence intervals, if it's calibrated
to more data points than the Mocha trial?

These large uncertainty intervals were driven by the calibration uncertainty while reproducing pre-
 intervention prevalence. After the various methodological changes implemented (addition of EHTs,
 calibration of pyrethroid-only ITNs to EHT data), the model for the Protopopoff trial does not have much
 larger uncertainty intervals anymore (see Figure 2).

- Line 351– please give a bit more detail about trial location, timing, etc

We have now included a table summarizing the main characteristics of the various EHT used:

	Experimental hut trial	Location	Year	Mosquito species	Pyrethroid-only ITNs	New-generation ITNs	Durability
Assenga et al. ⁵⁰	Tanzania (Ifakara, Morogoro)	Sept-Oct 2023	Anopheles arabiensis	MAGNet (alpha-cypermethrin)	Interceptor G2, Olyset Plus	Unwashed and 20x washed
Kibondo et al. ⁵¹	Tanzania (Ifakara, Morogoro)	Feb-Mar 2021	Anopheles arabiensis	Interceptor (alpha-cypermethrin)	Interceptor G2	Unwashed and 20x washed
BIT080 ⁵³	Tanzania (Ifakara, Morogoro)	Apr-May 2023	Anopheles arabiensis	MiraNet (alpha-cypermethrin)	Interceptor G2	Unwashed and 20x washed
BIT055	Tanzania (Ifakara, Morogoro)	Nov-Dec 2020	Anopheles arabiensis	Olyset (permethrin)	Olyset Plus	Unwashed and 20x washed
Odufuwa et al. ⁵²	Tanzania (Bagamoyo, Pwani)	Sept-Oct 2020	Anopheles arabiensis	PermaNet 2.0 (deltamethrin)	Olyset Plus	Unwashed and 20x washed
Martin et al. ⁵⁴	Tanzania (Magu, Mwanza)	May 2020-Dec 2022	Anopheles gambiae s.l.	Interceptor (alpha-cypermethrin)	Interceptor G2 Olyset Plus	New and field-aged ITNs after 36 months
Martin et al. ⁵⁴	Tanzania (Magu, Mwanza)	May 2020-Dec 2022	Anopheles funestus	Interceptor (alpha-cypermethrin)	Interceptor G2 Olyset Plus	New and field-aged ITNs after 36 months
Assenga et al. ⁵⁰	Côte d’Ivoire	May-July 2023	Anopheles gambiae s.l.	MAGNet (alpha-cypermethrin)	Interceptor G2, Olyset Plus	Unwashed and 20x washed
Nguessan et al. ⁵⁵	Benin (Cové, Zou)	Apr-July 2015	Anopheles gambiae s.l.	Interceptor (alpha-cypermethrin)	Interceptor G2	Unwashed and 20x washed
Sovegnon et al. ⁵⁶	Benin (Za-Kpota, Zou)	Jul-Aug 2021 and Oct-Nov 2021	Anopheles gambiae s.l.	Interceptor (alpha-cypermethrin)	Interceptor G2	New and field-aged ITNs after 24 months

- Line 365: I would make clear in the limitations/discussion that these results are only for *An.*
*arabiensis*, not only list it in the methods

Thank you for this remark. Our updated analysis now also includes one trials with results on *An.*
*funestus*, as well as three trials with *Anopheles gambiae s.l.* in West Africa. We have also added the
following limitation in the Discussion section:

*First, we estimated ITN entomological impact using EHTs from three countries: in Tanzania, results*
*were obtained for Anopheles arabiensis in five trials, and Anopheles funestus in one trial, in Benin and*
*Côte d'Ivoire, the results were obtained for Anopheles gambiae s.l. in three trials. Therefore, caution*
*should be exercised when extrapolating to other countries or vector species.*

- Line 398: what is kappa, and how does it relate to your other two killing params? Why is it there?

Many thanks for this remark. kappa represents the increase in killing associated with the intervention
and it is directly fitted during the inference procedure. It is implicitly included in the formula for the pre-
prandial killing effect phi, which is the parameter reported in the main results and used for subsequent
modelling. We have updated the text in the methods to improve clarity:

*A related indicator is the pre-prandial killing effect ϕ_i , namely the reduction in the probability p_{S_i} of*
*surviving before biting (where $p_{S_i} = 1 - p_{UD_i}$). The pre-prandial killing effect can be obtained via the*
*following formula (which can be sampled while fitting the Bayesian model):*

$$\phi_i = 1 - \frac{p_{S_i}}{p_{S_0}}$$

*This can be rewritten as*

$$p_{S_i} = p_{S_0}(1 - \phi_i)$$

*When $\phi_i = 0$, the intervention has no effect on the probability that mosquitoes survive, and when $\phi_i =$*
*1, all mosquitoes die and none can feed the protected host. **Because p_{S_i} is a function of α_i and μ_i ,***
***which themselves depend on π_i and κ_i , the pre-prandial killing effect ϕ_i is itself a function of both π_i***
***and κ_i .** It is required for subsequent mathematical modelling with OpenMalaria and will be used in the*
*model to represent the potential of the intervention to kill mosquitoes before feeding.*

- Table 1: a diagram/decision tree would be helpful to understand how these parameters interact

Thank you for this suggestion. The following figure was added within the Methods section:

*Figure 3. Schematic representation of the model used for statistical estimation of entomological effects. UA: unfed alive, UD:*
 *unfed dead, F: fed, FD: fed dead, FA: fed alive. The left panel (blue) represents the possible outcomes for a vector in presence of*
 *an unprotected individual in the control arm (untreated net). The right panel (red) represents the possible outcomes for a vector*
 *in presence of an individual protected by an ITN.*

- Line 434: How long is a time step?

The time step used is one day (or one night). This is now included in the unit column in the table with
 the parameter descriptions. Additionally, because in this model all parameters do not vary with time, we
 have now removed the time subscript t from most equations to improve readability.

- Table 2: why is the non-intervention counterfactual EIR different between trial arms?

This is because we calibrate the EIR to best reproduce the pre-intervention prevalence in each trial arm.
 Because pre-intervention prevalence is not equal in all arms (and most instances, pre-intervention ITN
 coverage also differs between arms), the best fitting EIR also differs between arms.

We added a sentence in the methods section to make this point clearer:

***Resulting EIR estimates may vary across trial arms due to differing baseline prevalence and***
 ***intervention coverages.***

**Reviewer #2 (Remarks on code availability):**

**I did a very brief scan of the GitHub repo, and it looks well-formatted and tidy. I know this takes a lot**
 **of work to maintain, and it's much appreciated. I would recommend adding a folder with all the data**
 **shown in figures, and a script allowing users to replicate the plots precisely.**

We have now included the data as part of the GitHub repository, and not as supplementary material
 anymore, for increased readability.

Reviewer #3 (Remarks to the Author):

This paper brings about an interesting solution for a very relevant problem: How to optimize the
deployment of ITNs from an explicitly epidemiological perspective, considering virtually all relevant
practical parameters associated with this important public-policy intervention. It combines the
analysis of experimental hut trials (EHTs), which were validated against Randomized Clinical Trials
(RCTs) and used for parameterization of a previously existing malaria-epidemiological modelling
framework (OpenMalaria).

The article is generally a good read with a clear and concise message. There are a few tweaks
regarding abbreviations that appear in the text without previous explanation. Even though some of
them are obvious for those familiar with the field, it is important to make the text accessible to
scientists outside the domain, as well. E.g., 'EHT' appears for the first time in the text on line 92; LLIN,
line 98. I suggest a review of accessibility of abbreviations and further domain-specific terminology
throughout the text (with the recognition that Textbox 1 covers a great deal of this issue already).

Thank you for this remark. EHT is now defined in line 48 and ITN in line 1. We have also revised the
wording of the Textbox, following the suggestion of Reviewer 2.

The method of EHT analysis, using a Bayesian framework is truly fascinating. It is a pity that the
approach is not explicitly mentioned in the Results section, which leaves the reader skeptical about
the validity of the analysis work. Highlighting Bayesian metrics in the Results section could be one way
to avoid unnecessary confusion.

Thank you for this remark. We added a mention of the Bayesian framework in the caption of Figure 1
and have reformulated the first paragraph of the results section to better highlight the Bayesian model
as follows:

***With a Bayesian model adapted from Denz et al.⁴⁰ (see Methods section) we estimated four indicators:***
***reduction in host availability (reflecting the deterrent properties of the nets), pre-prandial killing effect***
***(referring to mosquito mortality before biting), post-prandial killing effect (referring to mosquito***
***mortality after biting) and overall entomological efficacy (measured as the reduction in vectorial capacity***
***and is a combination of the previous three indicators). The indicators were estimated for Interceptor[®]***
***G2, Olyset[®] Plus and five pyrethroid-only ITNs (Interceptor, MiraNet, MAGNet, Olyset, PermaNet 2.0)***
***using data from eight EHTs conducted in Tanzania⁴⁸⁻⁵¹, Côte d'Ivoire⁴⁸ and Benin^{52,53} (cf. Figure 1, with***
***all numerical outputs in Supplementary Table 1).***

We also made clearer mention of Bayesian uncertainty quantification when commenting the estimates:

***In all studies but two (Assenga et al.⁴⁸, Tanzania data and Martin et al.⁵¹ for *An. funestus* mosquitoes),***
***the difference in vectorial capacity reduction between next-generation ITNs and pyrethroid-only ITNs***
***was positive in more than 95% of the simulations, indicating the superiority of next-generation ITNs in***
***terms of entomological efficacy (see Supplementary table 3.2). In the studies which included both***
***types of next-generation net within the same trial in Tanzania (Assenga et al.⁴⁸ and Martin et al.⁵¹),***
***the difference in vectorial capacity reduction estimates between Interceptor[®] G2 and Olyset[®] Plus had***
***a 95% credible interval containing the value 0. This indicates that, considering the statistical***
***uncertainty, we could not conclude that there was a difference in impact between these two products.***

***In the EHT conducted in Côte d'Ivoire⁴⁸, however, the interval did not contain 0 and a higher***
***entomological efficacy was measured for Interceptor G2.***

Additionally, as per the suggestion of Reviewer 2, the methods section now includes a diagram
describing the statistical model used, to improve clarity.

***That being said, one of my main concerns is the limited number of trials included in the analysis and***
***even smaller number brought into the model validation. I failed to find the right justification for the***
***small number of trials in the text and, while I have suspicions on the reasons why, I would like to***
***know that explicitly from the authors: Are there any other opportunities for an improved sampling of***
***studies? It seems a bit of a stretch to base some of much of a potentially immensely useful set of***
***recommendations on just two validations, one of which of very questionable fit (Protopopoff et al.***
***2018, 2023). To me, this is a major point that needs clear justification, consideration of impact on the***
***results and associated recommendations, and a proposal of a way forward for a robust policy***
***recommendation. There is one very strong claim on line 263, which – to me – needs substantial more***
***backing on data.***

Thank you for raising this important point. We have now expanded the number of trials included in the
manuscript.

For the estimation of entomological efficacy with the Bayesian model, we added four EHTs:
- Martin et al. 2024: conducted in Tanzania, alongside the Mosha et al. RCT, and comparing
pyrethroid-only ITNs with Interceptor G2 and OlysetPlus
- Nguessan et al. 2016: conducted in Benin and comparing pyrethroid-only ITNs with Interceptor
G2
- Sovegnon et al. 2024: conducted in Benin and comparing pyrethroid-only ITNs with Interceptor
G2
- Assenga et al. 2025: we now also include the trial data from Côte d'Ivoire, which compares
pyrethroid-only ITNs with Interceptor G2 and OlysetPlus.

For validation of the prevalence estimates at the population level, we added 2 RCTs:
- Staedke et al. 2020: conducted in Uganda, and comparing pyrethroid-only ITNs with PBO-
pyrethroid ITNs)
- Accrombessi et al. 2023: conducted in Benin, and comparing pyrethroid-only ITNs with
chlorfenapyr-pyrethroid ITNs.

To our knowledge, our analysis now includes all 4 RCTs comparing pyrethroid-only ITNs with Interceptor
G2 and/or PBO-pyrethroid ITNs available in the scientific literature (and meeting the inclusion criteria of
Sherrard-Smith et al. 2022)

We also edited the text in the discussion to nuance our claims:

***At the population level, we validated our dynamical model's ability to reproduce the reduction in***
***malaria prevalence observed in four RCTs conducted in Tanzania, Uganda, and Benin. These trials***
***consistently showed greater effectiveness of next-generation ITNs compared to pyrethroid-only nets,***
***though the duration of protection varied substantially across net types and settings. Despite some***

**discrepancies, our model reproduced the overall magnitude of the impact of next-generation ITNs on**
**prevalence and captured much of the associated uncertainty. When considering the effect sizes**
**between new-generation and pyrethroid-only ITNs and the relative comparison between pre- and**
**post-intervention levels, substantial variability was observed in both the observed and modelled**
**values within and across trials. Although the model did not capture perfectly the observed**
**heterogeneity for these two indicators, there was agreement in the general trends. Overall, this**
**evidence from RCTs is in line with the results observed the effectiveness pilots conducted in Burkina Faso,**
**Mozambique, and Rwanda, that also observed a superior effectiveness of chlorfenapyr and PBO nets**
**relative to pyrethroid ones for at least two years after distribution¹⁶. Less clear results were observed in**
**Nigeria where diverging trends in prevalence and incidence could be attributed to data quality issues,**
**confounding interventions and external factors, and where ITNs usage was low¹⁶. . Therefore, our**
**updated parameterization of the OpenMalaria model, which has been tested against RCT data and**
**demonstrates promising alignment especially for prevalence outcomes, can now be used to predict**
**potential ITN impact in places where there have not been RCTs. It can also be used for subnational**
**tailoring of interventions and/or country-wide predictions of the impact of malaria interventions^{35,58}.**

We also added this paragraph in the limitations section of the discussion:

**Second, despite an overall good agreement between observations and model predictions, there were**
**divergences in some instances, notably in Protopopoff et al.^{54,55}, where very high post-intervention**
**prevalence was reported towards the end of the trial, or in Accrombessi et al.^{14,15}, where the model**
**underestimated the impact of pyrethroid-only ITNs at the beginning and at the end of the trial.**
**Previous modelling attempts to reproduce these two trials with another malaria model faced similar**
**issues and outcomes^{38,42}. In Protopopoff et al.^{54,55}, some observations reported prevalence above 68%**
**among children under 15 years old, reaching as high as 81% in one arm: it is likely that the occurrence**
**of such high prevalence is due to external factors not included in the model. Mathematical models are**
**simplified representations of reality, which is far more complex than can be perfectly predicted. It is**
**therefore important to remember that, while such models can be useful at predicting general patterns**
**and trends, they are not necessarily effective for forecasting precise future outcomes.**

**A separate point, which does not pertain to the immediate epidemiological effect of ITN deployment,**
**but is of potentially disastrous impact, is the role of resistance evolution of the vector population to**
**the insecticides in consideration. This needs to be discussed and the work done in this area**
**considered, as well. The strategies of ITN deployment are not just a short-term problem, and**
**resistance can be a major driver of failure in the medium term. One particular aspect I would like to**
**see discussed is the interplay between insecticide resistance and the cascade of effectiveness.**

Many thanks for this remark. We have added the following paragraph in the Limitations section of the
Discussion:

**Finally, our model and associated tool do not incorporate considerations related to the management**
**of insecticide resistance spread. Insecticide resistance is constantly evolving, even as a consequence of**
**successful ITN interventions⁶⁵, and it can threaten the public health impact of ITNs. In our cascade,**
**resistance is implicitly included in the “entomological efficacy”: this step reflects the properties of the**

*ITNs in terms of killing and feeding inhibition under the resistance levels at the time and location*
*where the EHTs were conducted. Building on other modelling approaches that include a relationship*
*with bioassay mortality⁴³ or a dynamical representation of resistance spread⁶⁷ could help to*
*extrapolate predictions more accurately to new geographies or the future. While this is beyond the*
*scope of the current analysis, our model could be applied to more EHT data as they get generated, to*
*follow how entomological efficacy estimates change in reaction to resistance spread.*

*All in all, this paper proposes a tantalizing approach to improving the comparison and deployment of*
*ITNs: A theoretical solution to a very practical and real problem. Yet, “extraordinary claims require*
*extraordinary evidence”. In order to recommend the acceptance of this paper, I need see the limited*
*sample of studies addressed in the text. My recommendation is to include major revisions of the*
*included studies to validate the very powerful claim this paper makes.*

Thank you for your constructive feedback and for recognizing the value of our analysis. We hope that
the revised manuscript, which now includes many more studies for both calibration and validation,
provides solid support for our conclusions.

October 20th, 2025

Dear Editor,

We thank the Editor and the reviewers for their thorough read of our work and their appreciation of our revised manuscript. We have edited the article to address all remaining comments and provided a detailed answer to each of them below. All edits in the text are depicted in **bold** in the responses below.

We thank you very much for your time and consideration.

Sincerely,

Clara Champagne, on behalf of all authors

REVIEWERS' COMMENTS

Reviewer #1 (Remarks to the Author):

Thanks for the revised draft and updated dashboard to review.

I think the work is improved for this revision. The inclusion of the additional EHTs is great and also the validation effort against additional studies increasing the potential applicability of the dashboard.

Many thanks for this positive appreciation of our work.

One question of interest: When simulating the trials for Tanzania (Mosha), did you match the replacement of ITNs throughout as was explored in Churcher et al 2024? This could have made a substantial impact and it would be interesting to hear how you addressed this (I could not see it in the supplement)

In the trial, Mosha et al. reported participants' use of both study nets and any LLINs. In the simulations conducted, we assumed that all individuals initially possessed study nets, which were then lost over time according to a decay function derived from the observed data. The use of "any LLINs" remained relatively stable throughout the trial period so reduction in user and community level malaria were due to the decline of the study nets (further explored in Lukole et al 2025).

By contrast, Churcher et al. assumed that individuals maintained a constant overall ITN usage but switched between standard LLINs and next-generation nets.

These differences, alongside differences in model structure and entomological model calibration could explain the differences observed between our work and the one by Churcher et al.

We made our assumption more explicit in the methods section:

*Functional survival is included by reducing the effective ITN usage over time. **It is assumed that all individuals initially possessed study nets, which were then lost over time according to a decay function derived from the observed data. The decay distribution was assumed to be a Weibull function with half-life and shape parameter fitted to RCT data on ITN usage as explained in the Supplementary material (section 10), assuming half-life cannot exceed 3 years.***

Only minor things:

line 182 - spelling degradation should be degradation

done

Please check supplementary info are referred to in order

The supplementary information is now numbered in order of appearance in the main text.

Might be worth flagging the 25m point for the Protopopoff study that is missed in the pyrethroid only ITN arm.

We edited the text of the results as follows:

*For pyrethroid-only ITNs, although the models did reproduce general trends, seven out of 17 observed prevalence points were not captured by the model estimates. In Moshia et al. ^{11,12}, two observations were slightly missed. In Protopopoff et al. ^{53,54}, three observations with high prevalence toward the end of the trial were missed, **in particular the observations at 21 and 28 months**. In Accrombessi et al. ^{13,14}, the impact of pyrethroid-only ITNs was underestimated at the beginning and at the end of the post-intervention period.*

Reviewer #1 (Remarks on code availability):

The github is well organised and flagged appropriately throughout the script and code repository. The processed data is a useful resource.

I have not run the full analysis due to time limitations - but it reads well for the spot checks I tried.

Reviewer #2 (Remarks to the Author):

I commend the authors for the remarkable work that went into revising this manuscript. I now believe it to be much stronger, and can recommend it for publication.

Many thanks for this positive appreciation of our work.

My only note is to ask for more detail on previous work in the field. On lines 83-90, the authors state that effectiveness cascades for vector control have been presented, but rarely formally quantified. In addition to a few site-specific, data-driven examples (refs 30-32), they cite one modeling paper that approaches this same problem (ref 33). However, they do not discuss how their modeling analysis is different from this previous study. I am aware of at least one other modeling analysis, from the 2022 World Malaria Report, which approaches a very similar question. I'd like the authors to please extend this paragraph to briefly discuss these two studies, and any other related modeling work in the field, and state clearly how their work expands upon or diverges from these prior analyses.

Thank you for this remark and for pointing to us other modeling work that we had not mentioned. We have expanded the paragraph in the introduction as follows:

For vector control, and ITNs in particular, while conceptual cascades have been presented in the literature^{23,29}, they have rarely been formally quantified³⁰⁻³² due to the difficulty of disentangling factors that are intrinsically intertwined in real-life settings. Mathematical models of malaria transmission offer the possibility to disentangle each factor and quantify their relative contribution to the overall ITN effectiveness. Two previous modelling studies (Briet et al.³³ and Bertozzi-Villa et al.¹⁷, including the related analysis presented in the 2022 World Malaria report (<https://www.who.int/teams/global-malaria-programme/reports/world-malaria-report-2022>)) explored this topic. Both studies highlighted the critical importance of operational factors, such as access, use and durability. However, except for one net PBO-pyrethroid net included in Briet et al.³³, both studies mostly focused on pyrethroid-only ITNs and neither study directly incorporated human and vector activity rhythms in their quantifications. Additionally, both studies identified a substantial variability across countries and settings. There is therefore a need for additional tools that can explicitly quantify the relative contributions of the factors affecting the effectiveness of next-generation ITNs in various contexts. This would enable decision-makers

to assess which products are likely to perform best in their specific settings.

Reviewer #2 (Remarks on code availability):

The code repository continues to look tidy and well-organized. The authors state that they have included a folder with all the data shown in the figures, and a script allowing users to replicate the plots precisely, but I was not able to find it. Can the authors please label this directory more clearly, and directly reference it in the code availability section?

The EHT input data is available here:

https://github.com/SwissTPH/ITNcascades/tree/main/EHT_fit/processed_data.

Figures 1 and 4 can be reproduced based on this and the code available in

https://github.com/SwissTPH/ITNcascades/tree/main/EHT_fit

The results of the dynamical model simulations are provided here:

https://github.com/SwissTPH/ITNcascades/tree/main/RCT_validation/csv_inputs

Figures 2 and 3 can be reproduced using these inputs and the code in

https://github.com/SwissTPH/ITNcascades/blob/main/RCT_validation/5_validation_summary_figures.R

In the data availability statement, we have edited the link to point directly to the folder where data is contained.

In the GitHub repository, we have added a section in the readme entitled “reproducing the manuscript’s figures”.

Figures 1 and 4 can be reproduced based on the data and this code.

Figures 2 and 3 can be reproduced using the dynamical model simulations and this code.

Reviewer #3 (Remarks to the Author):

Thank you for addressing, not only my concerns, but all of the reviewers' points. I am now satisfied with the quality of the work and the evidence backing its claims. My recommendation is for acceptance.

Many thanks for this positive appreciation of our work.

Reviewer comments: NCOMMS-25-01112

This is an excellent contribution. The draft is well written, presented and considered and highly valuable to the malaria community. The work is novel in that it provides a flexible platform to consider cascading loss of impacts from the delivery of ITNs of two classes that are in regular use across malaria endemic regions. It is particularly useful to be able to consider the loss in impact due to the product in contrast to that due to operational challenges such as ITN access, or usage and exposure risks through human biting or mosquito activity patterns. I support its publication in Nature Communications and believe it fits the journal well.

I have only some minor suggestions / typo highlights to offer. But please consider the suggestions for the app which might make it more translatable

Have you tested the process elsewhere where other EHTs used? It might be a worthwhile exercise to confirm results are not substantially different? Appreciate that you would need to validate locally but perhaps possible in Benin to at least provide 1 alternative species.

Feedback on the app

At the moment, the first thing we see on the dashboard are the figures, which need some explanation to interpret. Could you make the default page the "About" tab, and add a "Definitions" tab to this page adjacent to the About tab. It would be excellent too to have an "Interpreting the outcome" explainer to support users.

The details for the various parameters are noted within the Assumption tab, I think it could be useful to tell the user this. Under the Parameters title on the left of the screen, could you add; "For definitions and assumptions on values please see Assumptions tab" or something similar.

For attrition, it could be really useful to flag this tool: ITN Quantification that would allow countries (outside of Tanzania) to gain information on their location.

Could you have a button that automatically allows you to fill the same information for the two ITNs so that the user can ensure no mistake in typing causes a comparison that is unmatched? I appreciate there are measurable differences in the EHTs that would be retained, but might be useful for decisions makers.

Linked to the only query above, could you add other experimental hut trial estimations from West Africa? To the five described by BIT103⁴⁸, BIT055, BIT080⁴⁹, Odufuwa⁵⁰, and Kibondo⁵¹

The Activity Patterns for the two ITNs are identical (which might be what you desire but flagging in case as these are separate tabs)

Minor considerations

Introduction

Perhaps in the introduction, where you say:

In 2023, overall access to ITNs was estimated at around 60% in Africa, with only five countries on the continent achieving the target of 80% of the population sleeping under an ITN ^{4,18}.

Add

In 2023, overall access to ITNs was estimated at around 60% in Africa, with only five **of XX** countries **with endemic malaria** on the continent achieving the target of 80% of the population sleeping under an ITN ^{4,18}.

As this will give better context.

Perhaps in the introduction, where you say:

Various indicators exist to monitor insecticide resistance, ITN use, ITN durability and in-bed exposure (see Textbox 1) but these indicators are measured independently of each other and therefore do not provide information on the relative contribution of each factor to the overall public health impact of ITNs.

Add

Various indicators exist to monitor insecticide resistance, ITN use, ITN durability and in-bed exposure (see Textbox 1) but these indicators are **usually** measured independently of each other and therefore do not provide information on the relative contribution of each factor to the overall public health impact of ITNs.

As there are examples of studies that try to generate all these information, see xxxx.

Where you state

In particular, EHT data has been used to estimate the entomological efficacy of various vector control tools and hence parameterize mathematical models of malaria ⁴¹⁻⁴⁴.

It is probably worth adding two key publications in this space

Nash 2021

Challenger 2023

Results

A typo to correct here:

Entomological efficacy was quantified in four indicators: reduction in host availability (reflecting the deterrent properties of the nets), pre-prandial killing effect (referring to mosquito mortality before biting), post-prandial killing effect (referring to mosquito mortality after biting) and overall reduction in vectorial capacity (representing **to** the potential for a given mosquito population to transmit malaria by combining the previous three indicators).

Methods

Where you mention “unfed-alive, unfed-dead, fed-alive and fed-dead” perhaps add the UA, UD, FA, FD acronyms that align with the supplementary data files.

There is a reference to Appendix 4 but I cannot find this document (potentially needs updating – I think to Supplementary Table 1), similarly I think the ref to Appendix 2 should be Supplementary Table 2? Think same throughout – presumably the journal instructions can advise.

Typo: all notations are indicated in **Error! Reference source not found.** (*needs a space*)

For the units, I think time is 1 day (or 1 night...) perhaps specify this if so in the Tables of parameters time^{-1} *where unit time is 1 day*

Just need to double check the order of Tables/Figs mentioned is as per article... I think Table 2 is referenced after Table 3 – very minor point!!

Discussion

The main findings are well presented and discussed, the only suggestions I have here is:

The only species included in the analysis is *arabiensis* – could you add a limitation to this effect. I appreciate that were the methodology applied to other EHTs you could generate the entomological efficacy reductions to build the framework, but at this stage, in the app particularly, the estimates are from the 5 EHTs that consider *arabiensis* only.

Congratulations on a great contribution.